



# Black carbon modelling in urban areas: investigating the influence of resuspension and non-exhaust emissions in streets using the Street-in-Grid (SinG) model

Lya Lugon[1,2], Jérémy Vigneron[3], Christophe Debert[3], Olivier Chrétien[2], and Karine Sartelet[1]

[1]CEREA, Joint Laboratory École des Ponts ParisTech/EDF R&D, Université Paris-Est, 77455 Champs-sur-Marne, France
[2]Paris City, Departement of green spaces and environment, 103 Avenue de France, France
[3]Airparif, France

**Correspondence:** Lya Lugon (lya.lugon@enpc.fr), Karine Sartelet (karine.sartelet@enpc.fr)

**Abstract.**

Black carbon (BC) is a primary and inert pollutant often used as a traffic tracer. Even though its concentrations are generally low at regional scale, BC presents very high concentrations in streets (local scale), potentially causing important effects on human health and environment. modelling studies of BC concentrations usually underestimate BC concentrations, because of uncertainties in both emissions and modelling. Both exhaust and non-exhaust traffic emissions present uncertainties, but those on non-exhaust emissions, such as tyre, brake and road wear and particle resuspension, are particularly high. In terms of modelling, the street models do not always consider the two-way interactions between the local and regional scales, i.e. the influence of the high BC concentrations observed in streets on the urban background concentrations, which can enhance the BC concentrations in streets. This study uses the multi-scale Street-in-Grid model (SinG) to simulate BC concentrations in a Paris suburb street-network, taking into account the two-way coupling between local and regional scales. The BC concentrations in streets proved to have an important influence on urban background concentrations. The two-way dynamic coupling leads to an increase in BC concentrations in large streets with high traffic emissions (with a maximal increase of about 48%), and a decrease in narrow streets with low traffic emissions and low BC concentrations (with a maximal decrease of about 50%).

A new approach to estimate particle resuspension in streets is implemented, strictly respecting the mass balance on the street surface. The resuspension rate is calculated from the available deposited mass on the street surface, which is estimated based on a particle deposition and wash-off parametrizations adapted to street-canyon geometries. The simulations show that particle resuspension presents a low contribution to black carbon concentrations, as the deposited mass is not significant enough to justify high resuspension rates.

Non-exhaust emission, such as brake and tyre and road wear, largely contribute to BC emissions, with a contribution equivalent to exhaust emissions. Here, emission factors of tyre, brake and road wear are calculated based on the literature, and a sensitivity analysis of these emission factors on BC concentrations in streets is performed. The model to measurement comparison shows that tyre-emission factors usually used in Europe are probably under-estimated, and tyre-emission factors coherent with some studies of the literature and the comparison performed here are proposed.





# 1 Introduction

Black carbon (BC) is a primary and chemically inert atmospheric pollutant, compound of $PM_{2.5}$ (particulate matter of diameter lower than 2.5 $\mu$m). BC background concentrations in urban areas are generally quite low, but they can reach high values in streets. The Parisian air-quality agency (AIRPARIF) performed a chemical speciation of particulate matter between September 2009 and September 2010, and observed that BC concentration represented approximately 4% of $PM_{2.5}$ concentrations in rural areas, 10% in urban background, reaching 27% in a street with high traffic (AIRPARIF, 2012). Because BC is mainly emitted

by traffic, it is often used as a traffic pollutant tracer (Invernizzi et al., 2011; de Miranda et al., 2019). A special attention is given to BC because of its potential impacts on human health: recent studies showed a strong correlation between BC concentrations and the occurrence of respiratory and cardiovascular problems, others indicated an alteration on fetus development (Jansen et al., 2005; Highwood and Kinnersley, 2006; Janssen et al., 2011; Dons et al., 2012; Zhang et al., 2019a, b). BC is also responsible for environmental impacts, such as visibility reduction (Tao et al., 2009; Chen et al., 2016; Li et al., 2019) and

radiative effects (Jacobson, 2001; Chung and Seinfeld, 2005; Tripathi et al., 2005; Ramachandran and Kedia, 2010).

Although BC concentration is high in urban streets, BC concentration and its dispersion are often not modelled at the street scale. Most of air-quality studies regarding BC employ statistical regression techniques, that estimate BC concentrations over a street network based on meteorological data and punctual observations (Richmond-Bryant et al., 2009; Awad et al., 2017; Van den Bossche et al., 2018; Sanchez et al., 2018; Van den Hove et al., 2019; Boniardi et al., 2019; Liu et al., 2019;

Jones et al., 2020). When street or street-network models are used, emission data are often modified and/or are coupled to observational data to improve modelled concentrations. For example, Brasseur et al. (2015) used a street model over two street canyons in Brussels. To achieve the high BC concentrations observed in the streets, a correction factor of 3.0 was applied to traffic emissions calculated with COPERT IV, a database that provides traffic emissions factors for European vehicles (Ntziachristos and Samaras, 2018). This under-estimation is also suggested in studies using more complex local-scale models

for BC dispersion. For example, Tong et al. (2011) simulated BC concentrations using the Comprehensive Turbulent Aerosol Dynamics and Gas Chemistry (CTAG) in an urban region at South Bronx, New York. A good correlation between observed and simulated BC concentrations was reached combining field measurements and numerical simulations and increasing by 15% the number of heavy-duty diesel vehicles in the morning peaks (and consequently increasing BC emissions). The corrections applied to BC emissions in these studies suggest an under-estimation of BC emissions in models.

Uncertainties in BC concentrations modelled with street-network models are partly due to uncertainties in traffic emissions. Traffic emissions are classified as exhaust emissions, and non-exhaust emissions, from tyre, brake and road wear and particle resuspension. On one hand, exhaust emissions are relatively well-known, as they are characterized in laboratory controlled conditions. In Europe, exhaust emission factors are nowadays determined according to the vehicle technology and fuel, providing realistic emission factors for divers vehicle fleet, as detailed in the EMEP guidelines (Ntziachristos and Samaras, 2018). Emis-

sion factors are often determined for regulated pollutants, such as $PM_{10}$ and $PM_{2.5}$. However, Ntziachristos and Samaras (2018) also provides information about BC emissions, by providing a speciation of $PM_{2.5}$ in the form of $BC/PM_{2.5}$ ratios, which vary depending on the vehicle category (light-duty vehicles LDV or heavy-duty vehicles HDV), fuel and regulatory standard of





the vehicle manufacturing (Euro norm), as presented in Table 3.91 of Ntziachristos and Samaras (2018). The uncertainties of the BC/PM$_{2.5}$ ratios depend on the vehicle category. For LDV diesel vehicles, which present the highest BC/PM$_{2.5}$ ratios, the
uncertainties of BC/PM$_{2.5}$ ratios are quite low, ranging from 5 to 10%. For diesel vehicles equipped with particle filter, the uncertainties are high (about 50%), but with a lower BC/PM$_{2.5}$ ratio than other diesel vehicles, and also with a lower PM$_{2.5}$ emission rate.

On the other hand, non-exhaust emissions are still not well-known in the literature, and they are not included in regulatory emissions legislation. Emission factors are greatly variable and associated to large uncertainties. Non-exhaust emissions consist
of particles emitted from vehicles operation and linked to tyre, brake and road wear and resuspension. Uncertainties in non-exhaust emissions can be explained by the difficulty to dissociate emissions from the different processes and by the great variability of: ($i$) tyre, brake and road constituents used in different locations; ($ii$) vehicles characteristics, such as weight, location of driving wheels, etc; ($iii$) vehicles operational conditions, such as vehicles speed and ambient temperature; and ($iv$) methodologies to determinate wear emission factors, i.e. direct measurements *in-situ*, with wind-tunnel experiments or
receptor-oriented methods (Boulter, 2005; Thorpe and Harrison, 2008). These uncertainties are reflected in the estimation of the contribution of non-exhaust emissions to PM concentration observed in the literature. According to Berdowski et al. (2002), non-exhaust emissions contribute to only 3.1% of PM$_{10}$ and 1.7% of PM$_{2.5}$ concentrations in European countries, and specifically in France to 2.2% of PM$_{10}$ and 1.1% of PM$_{2.5}$. Similarly, Dore et al. (2003) estimated that the contribution of non-exhaust emissions is low: they estimated that 80% of inhalable PM are emitted from road traffic, of which only 3% are non-
exhaust emissions. Differently, other studies estimated that the contribution of non-exhaust emissions is very high. Rauterberg-Wulff (1999) obtained tyre wear emission factors using wind-tunnel experiments in controlled conditions of the same order of magnitude than diesel exhaust emissions. Using tunnel measurement techniques, Lawrence et al. (2016) quantified PM$_{10}$ emission factors from exhaust and non-exhaust sources for a vehicle fleet. They estimated non-exhaust PM$_{10}$ emission factor to be 50% higher than the exhaust emission factor: 49% of PM$_{10}$ emissions were estimated to be non-exhaust emissions, 33%
to be exhaust emissions and 18% were considered as unexplained emissions; brake and road wear presented an important contribution to PM$_{10}$ emissions, with almost the same emission factor as petrol exhaust. Resuspension process had the higher emission factor (10.4 mg.vkm$^{-1}$ against 4.5 mg.vkm$^{-1}$ of petrol exhaust and 8.3 mg.vkm$^{-1}$ of diesel exhaust), followed by unexplained sources (7.2 mg.vkm$^{-1}$). Harrison et al. (2001) measured PM$_{10}$ and PM$_{2.5}$ at five sites in the United Kingdom in order to perform a correlation between emission sources and PM$_{10}$ concentrations. They estimated non-exhaust emissions
to be approximately of the same magnitude as exhaust emissions. The non-exhaust emissions can also be very important for secondary particle formation, especially regarding secondary organic aerosol (SOA). Khare et al. (2020) highlighted the importance of asphalt-related emissions in SOA formation, still absent from emission inventories. The annual estimation of asphalt-related SOA precursor emissions at urban scales is estimated to exceed those from motor vehicles. These studies show that non-exhaust emissions can be very important for air-quality, and that it is necessary to improve the identification source
techniques and the parametrizations to quantify non-exhaust emission rates.

Some models try to integrate vehicle operational conditions to estimate non-exhaust emission factors, even if based on empirical factors and simplified parametrizations. The HERMES model employs the non-exhaust emission factors proposed in





the EMEP guidelines (Ntziachristos and Boulter, 2016), which provide $PM_{10}$ wear emission rates for passenger cars, motor-cycles, LDV and HDV. A speed correction ratio is adopted for tyre and brake wear emission factors, ranging from 1.39 and

0.902 for tyre wear and 1.67 and 0.185 for brake wear. Specifically for HDV, $PM_{10}$ tyre and brake-wear emissions are cal-culated taking into account vehicle weight, represented by a load factor ranging from 0 to 1. HDV tyre-wear emissions also take into account the vehicle size by the number of axles. A detailed description of non-exhaust emission factors employed in the HERMES model is available in the EMEP guidelines (Ntziachristos and Boulter, 2016), and a brief summary is presented in section 3.2 of this paper. $PM_{10}$ emission rates from tyre, brake and road wear indicated in the EMEP guidelines are quite

low. Tyre wear emission rates are around 6 mg.vkm$^{-1}$ for LDV, and 18 mg.vkm$^{-1}$ for HDV, brake wear emission rates are around 8 mg.vkm$^{-1}$ for LDV, and 40 mg.vkm$^{-1}$ for HDV, and road wear emission rates equal to 7.5 mg.vkm$^{-1}$ for LDV and 38 mg.vkm$^{-1}$ for HDV. Differently, the NORTRIP model takes into account the vehicle speed to determine $PM_{10}$ tyre and road wear emission rates, but brake-wear emissions are supposed to be independent of vehicle speed. The NORTRIP model adopts a linear regression based on tyre and road wear emission rates observed at a reference speed of 70 km.h$^{-1}$. Road-wear

emissions also take into account the soil characteristics, as the pavement hardness. A detailed description of non-exhaust emis-sion factors employed in the NORTRIP model is available in Denby et al. (2013a). Quite similar tyre and brake wear rates are obtained using the EMEP guidelines and the NORTRIP model. However, the road-wear emissions calculated using the NORTRIP model with the soil characteristics employed by Thouron et al. (2018) in "Boulevard Alsace Lorraine" (a street East of Paris) are higher than those proposed in the EMEP guidelines, by a ratio 6.0. More details about tyre, brake and road-wear

emissions using the NORTRIP model and the EMEP guidelines emission factors are presented in section 4.

Other studies assume constant non-exhaust emission factors for separate sources, and they show a great variability of several orders of magnitude between emission factors for tyre wear. The bibliographic review presented in Boulter (2005) shows different tyre wear emission factors, ranging from 10 mg.vkm$^{-1}$ to 100000 mg.vkm$^{-1}$. Tyre wear emission factor of the order of 100 mg.vkm$^{-1}$ for passenger cars are observed in the literature: Luhana et al. (2004) measured an average tyre wear emission

rate for passenger cars of 97 mg.vkm$^{-1}$, and Baumann and Ismeier (1997) of 80 mg.vkm$^{-1}$. According to the bibliographic review presented in the US Environmental Agency (EPA) report (available at https://cfpub.epa.gov/si/si_public_file_download. cfm?p_download_id=525701, Table 3-1), other studies proposed tyre-wear emissions rates for LDV around 100 mg.vkm$^{-1}$, such as Gebbe (1997) (110 mg.vkm$^{-1}$), and Malmqvist (1983) (120 mg.vkm$^{-1}$). Higher LDV tyre wear rates, around 200 mg.vkm$^{-1}$, were estimated by Councell et al. (2004) and Baekken (1993), and 300 mg.vkm$^{-1}$ by Schuring and Clark (1988)

(ranging from 240mg.vkm$^{-1}$ to 360mg.vkm$^{-1}$). Tyre wear emission rates could be even higher according to the ambient and operational conditions: Park et al. (2018) investigated tyre wear particles generated in a laboratory under different tyre/road contact conditions and observed important variations in tyre wear emission factors and size distributions with the variation of road cornering conditions. At constant speed conditions (80 km.h$^{-1}$), the tyre wear emission rates obtained with a 2° tyre slip angle were about 300 times larger than those obtained with no tyre slip angle, increasing from 3.5 mg.km$^{-1}$ to 1110.8

mg.km$^{-1}$ per tyre. Considering a vehicle with 4 tyres, the tyre-wear emissions range from 14 mg.vkm$^{-1}$ to 4443.2 mg.vkm$^{-1}$ only with a 2° tyre slip angle variation.



Brake-wear emissions present a lower variability, according to the literature. The greater variations in $PM_{10}$ brake wear emission are observed by Abu-Allaban et al. (2003) using a receptor modelling technique, ranging from 0 to 80 mg.vkm$^{-1}$ for LDV and 0 to 610 mg.vkm$^{-1}$ for HDV. Other experimental studies estimate lower brake wear emission rates, with typical

values around 8 mg.vkm$^{-1}$ for LDV and 40 mg.vkm$^{-1}$ (Grigoratos and Martini, 2015; Denby et al., 2013a; Sanders et al., 2003; Ntziachristos and Boulter, 2016).

Road emission factors vary from 3.8 mg.vkm$^{-1}$ (Boulter, 2005) to 200 mg.vkm$^{-1}$ (Thouron et al., 2018), and they present the worst quality codes compared to other wear emission factor. The EMEP guidelines (Ntziachristos and Boulter, 2016) quantify the typical error associated to road wear emission factors to be between 50% to 300%, associated to the difficulties to

separate precisely the non-exhaust emission sources and the dependence of soil properties and vehicles operational conditions.

Among the non-exhaust emissions, particle resuspension is probably the process that presents the largest uncertainties. Divers studies estimate resuspension emission factors, often higher than exhaust emissions but with a variability of several orders of magnitude. Luhana et al. (2004) observed relatively low resuspension factors: about 0.8 mg.vkm$^{-1}$ for LDV and 14.4 mg.vkm$^{-1}$ for HDV. Note that the same study observed an important contribution of tyre wear, with an average emission

factor of 94 mg.vkm$^{-1}$ for passenger cars. Lawrence et al. (2016) determined the emission factors from for a typical vehicles fleet (composed of 92% of passenger cars) with wind-tunnel experiments. Resuspension presented the higher emission factor (10.4 mg.vkm$^{-1}$), in contrast with lower emission factors from combustion (4.5 mg.vkm$^{-1}$ for gasoline vehicles and 8.3 mg.vkm$^{-1}$ for diesel vehicles), tyre and brake wear (4.4 mg.vkm$^{-1}$), road wear (4.5 mg.vkm$^{-1}$), and unexplained sources (7.2 mg.vkm$^{-1}$). For a similar fleet, Pay et al. (2011) measured $PM_{10}$ resuspension factors in Berlin, and estimated them to

be 88 mg.vkm$^{-1}$ for LDV and 217 mg.vkm$^{-1}$ for HDV. These values were adopted in the the HERMES model to calculate particles resuspension in Spain (Pay et al., 2011) and led to a similar impact on $PM_{10}$ concentrations as exhaust emissions. Note that other non-exhaust emission factors used in the HERMES model are based on the EMEP guidelines, but no resuspension emission rate is defined by Ntziachristos and Boulter (2016) due to the great uncertainties observed in the literature. In the HERMES model, resuspension emission factors are the most important non-exhaust emission source, and no precision about

vehicle operational conditions and the available mass in surface are considered. Other studies obtained even higher resuspension factors: 10-1000 mg.vkm$^{-1}$ (Venkatram et al., 1999), 7600-8400 mg.vkm$^{-1}$ (Moosmüller et al., 1998), 40-780 mg.vkm$^{-1}$ for LDV and 230-7800 mg.vkm$^{-1}$ for HDV (Abu-Allaban et al., 2003), showing the huge uncertainties in particle resuspension factors.

Considerable difficulties to differentiate tyre, brake and road-wear emissions from resuspension are mentioned in experimen-

tal studies (Thorpe et al., 2007; Ntziachristos and Boulter, 2016; Beji et al., 2020), which indicate that these wear emissions factors may be even higher than expected and with different possible classification among sources. Due these difficulties, Beji et al. (2020) classified non-exhaust emissions as brake-wear emissions and tyre-road contact particles, grouping tyre and road wear and resuspension in the same source.

Resuspension emission factors can be employed in air-quality models (Pay et al., 2011), but besides the uncertainties in these

emission factors, the methodology does not necessarily respect mass balance on the street surface. If resuspension is implemented in the model using resuspension emission factors, a mass balance between total particle emissions, deposition, drainage





caused by rain (wash-off) and resuspension factors may not hold. However, this mass balance may be used to determine the available particle mass on the street surface that may be resuspended. The NORTRIP model (Denby et al., 2013a, b) computes particle resuspension based on a street surface mass balance, also integrating the particle wash-off. But the NORTRIP model

artificially assumes that tyre, brake and road-wear emissions are instantly deposited over the street surface, and only these sources are employed to calculate the particle mass on the street.

Beyond the uncertainties attached to PM non-exhaust emission factors, their chemical composition is still not well-known. Tyre-wear emissions contain a large fraction of BC, as BC represents 22%-30% of tyre weight (Thorpe and Harrison, 2008). Quite similarly, the BC fraction observed in tyre-wear emitted particles, is about 13%-19% (Kreider et al., 2010), 18% (Park

et al., 2017), and 15.3% (Ntziachristos and Boulter, 2016). Road and brake wear present lower BC fractions, with 1.06% ($\pm$ 50%) and 2.6% respectively according to Ntziachristos and Boulter (2016). More recently Lyu and Olofsson (2020) investigated BC emissions from disc brakes, and concluded that the BC fraction in $PM_1$ emitted from brake wear is higher than the BC fraction from combustion process. The BC fractions observed in $PM_1$ emitted by three different types of brake ranged from 20.7% to 72.4%, with an average value of 41.5%, depending on the surface conditions and graphite content of the brake

materials. These studies regarding non-exhaust emission characteristics emphasize that the knowledge on the BC emissions from tyre, brake and road wear is far from complete, and further studies are required.

Another aspect that may affect the under-estimation of BC concentrations in simulations is the one-way coupling approach usually employed in street-network or local-scale models. Street-network models often use prescribed background concentrations. Although the vertical mass transfer between the streets and the background influenced the concentrations in the streets,

its influence on the background concentrations is often neglected. The multi-scale model Street-in-Grid (SinG) combines the Model of Urban Network of Intersecting Canyons and Highways (MUNICH) (Lugon et al., 2020; Kim et al., 2018) for modelling street concentrations and Polaid3D for modelling background concentrations (Sartelet et al., 2007), allowing to simulate local and regional scale simultaneously. SinG performs a two-way coupling between regional and local scales, taking into account at each iteration the influence of the vertical mass transfer between the background and the streets on both the background

and the street concentrations. This two-way coupling between the streets and the background can be especially important for BC, as the concentrations observed in streets are considerably larger than BC concentrations in the urban background. Previous studies regarding gas-phase pollutants observed an important effect of the two-way coupling on $NO_2$, $NO$ and $NO_x$ concentrations in the streets, which may reach 60% in streets with high traffic emissions (Lugon et al., 2020; Kim et al., 2018).

This study presents simulations of BC concentrations in a Parisian suburb street-netwok using the model SinG. It investi-

gates ($i$) the influence of non-exhaust emissions on BC concentrations in the streets, presenting a new approach to estimate particle resuspension respecting mass conservation on the street surface; and ($ii$) the importance of a two-way coupling between regional and local scales by comparing BC concentrations in streets calculated by SinG and MUNICH. Emission factors of tyre, brake and road wear are calculated based on the literature and sensitivity tests are performed. Model to data comparisons are based on BC observations performed during the TRAFIPOLLU campaign (https://trimis.ec.europa.eu/project/

multiscale-modeling-traffic-pollutants-urban-area), in a street called "Boulevard Alsace Lorraine". The same measurement site was already used by Kim et al. (2018) for the evaluation of NOx and $NO_2$ concentrations simulated by SinG. Section 2





describes SinG, emphasizing the parametrizations of particles deposition, wash-off and resuspension. Section 3 summarizes simulations setup, at both regional and local scales. Section 4 describes the sensitivity analysis to estimate the importance of tyre, brake and road-wear emissions on BC concentrations, and section 5 presents the influence of the two-way coupling
between the regional and local scales on BC concentrations in the streets.

## 2   Model description

Street-in-Grid (SinG) is a multi-scale model that performs a dynamic two-way coupling between the street-network model MUNICH (Model of Urban Network of Intersecting Canyons and Highways) and the 3D chemistry-transport model Polair3D. As detailed in Lugon et al. (2020), this dynamic coupling between local and regional scales allows a direct interaction between
concentrations in the street network and those in the urban background: the mass transfer between the street and the background concentrations influence both the street and the background concentrations. Furthermore, SinG uses consistent chemical and physical parameterizations, such as the same chemical module and meteorological data, at both local and regional scales. SinG and MUNICH are described in Lugon et al. (2020) and Kim et al. (2018), and Polair3D is presented by Boutahar et al. (2004); Sartelet et al. (2007), all available in the Polyphemus platform (Mallet et al., 2007). The size distribution of BC is modeled
with a sectional approach, with diameters ranging typically from $10^{-3}$ $\mu$m and 10 $\mu$m. Because BC is an inert species, this study does not take into account chemical reactions, and only BC concentrations are modelled.

Particle resuspension is the non-exhaust emission process that presents the largest uncertainty and variability. Different studies in the literature propose constant resuspension emission factors (see section 1), but this methodology does not necessarily respect mass conservation on the street surface. This study presents a new methodology to calculate particle resuspension,
based on the respect of mass conservation on the street surface. At each time iteration, SinG calculates the total particle mass available on the street surface ($M_{dep}$), taking into account particle deposition ($Q_{dep}$), wash-off ($Q_{wash}$) and resuspension ($Q_{res}$) rates, by integrating the following equation:

$$\frac{dM_{dep}}{dt}\bigg|_{surf} = Q_{dep} - (Q_{wash} + Q_{res}). \tag{1}$$

As detailed in section 2.3, the parametrizations used to calculate particle resuspension are based on the NORTRIP model
(Denby et al., 2013a), but with an important difference: NORTRIP calculates the deposited mass assuming that all wear emissions are directly deposited, and deposition is not directly linked to the concentrations of particles in the street. Here, the concentrations of particles are computed with SinG, as detailed in Lugon et al. (2020), and particle deposition is computed from the concentrations, meaning that the time to deposit is taken into account (no instantaneous deposit), and particles from all origins (e.g. exhaust, non exhaust, particles transported from other sources) may deposit. The formulations used for deposition,
washout and resuspension in the street surface mass balance equation are now detailed.





## 2.1 Particle deposition

Particle and gas dry-deposition modelling follows the Cherin et al. (2015) approach, designed for street canyons. This parametrization calculates separately particle deposition over the different available surfaces in a street canyon, such as pavement area and building walls. Note that only the particle deposition on the street pavement is considered when computing the street-surface deposited mass, which is available for resuspension. A complete description of this approach is detailed in Cherin et al. (2015), with the computation of deposition velocities for gas and particulate species. The deposition mass rate $Q_{dep}$ is proportional to the deposition velocity $v_{dep}$ and the species concentration in the street $C_{sp}$. Particle deposition velocities varies with particle diameter, indicated by the size section $b$:

$$Q_{dep} = v_{dep,b} \times C_{sp,b}. \tag{2}$$

## 2.2 Particle wash-off

Particle wash-off over the street surface is computed using the concept of wash-off factor $f_{wash}$, as described in Denby et al. (2013a, b). In this parametrization, the particle mass rate drained by water $Q_{wash}$ is proportional to $f_{wash}$ and the deposited mass over the surface $M_{dep}$, as detailed in Equations (3) and (4):

$$f_{wash} = \frac{1}{\delta t} \left( 1 - exp \left( -h_{drain,eff} \frac{g_{road} - g_{road,min}}{g_{road,min}} \right) \right), \tag{3}$$

$$Q_{wash} = f_{wash} \times M_{dep}, \tag{4}$$

with $\delta_t$ the time step between two evaluations of $g_{road}$ (600 s here), $h_{drain,eff}$ the drainage efficiency parameter, $g_{road}$ the amount of water presents on the street surface [mm], $g_{road,min}$ the minimum water content to drainage process [mm]. Note that the drainage efficiency $h_{drain,eff}$ depends on particle properties. It can range from 0 to 1, depending on the species mixing state and solubility. As discussed in Denby et al. (2013a) and Vaze and Chiew (2002), salt should be well mixed and very soluble in water, thus having high drainage efficiency. Because dust and BC are insoluble species, and because they may not be well mixed with salts (Zhu et al., 2016; Majdi et al., 2020), their drainage efficiency should be poor. Here, $h_{drain,eff}$ is taken equal to 0.001, as proposed in Denby et al. (2013b); $g_{road}$ corresponds to the water rain calculated in each street and integrated in time from the meteorology, and $g_{road,min}$ is 0.5 mm, as in Denby et al. (2013b). Drainage is treated in the model as an instantaneous process, as performed by Denby et al. (2013a) and Denby et al. (2013b).

## 2.3 Particle resuspension

Particle resuspension is calculated based on a resuspension factor $f_{res}$, as proposed by Denby et al. (2013a) and employed in the NORTRIP model. This factor varies depending on the traffic flow characteristics, such as the LDV and HDV vehicle flow





and speed:

$$f_{res} = \sum_{v=1}^{2} N_v \left( \frac{u_v}{u_{ref(r)}} \right) f_{0,v}, \tag{5}$$

where $v$ indicates the vehicle type, $N_v$ the vehicle flow [veh.h$^{-1}$], $u_v$ the vehicle speed [km.h$^{-1}$], $u_{ref(r)}$ is the reference vehicle speed for the resuspension process [km.h$^{-1}$], and $f_{0,v}$ the reference mass fraction of the resuspension process [veh$^{-1}$]. As used in (Denby et al., 2013a) and Thouron et al. (2018), this study adopts $u_{ref(r)} = 50$km.h$^{-1}$, $f_{0,hdv} = 5 \times 10^{-5}$veh$^{-1}$ and $f_{0,ldv} = 5 \times 10^{-6}$veh$^{-1}$.

The resuspension rate $Q_{res}$ is then calculated as detailed in Equation (6), as a function of the deposited mass on the street 260 surface $M_{dep}$ [$\mu$g]:

$$Q_{res} = f_{res} \times M_{dep}. \tag{6}$$

### 2.4 Solving particle mass balance over the street surface

Using the formulations of wash-off and resuspension presented in Equations (4) and (6), the time evolution of the surface mass (Equation (1)) can be rewritten as:

$$\left. \frac{dM_{dep}}{dt} \right|_{surf} = Q_{dep} - M_{dep}(f_{wash} + f_{res}), \tag{7}$$

with an analytical solution:

$$M_{dep(t)} = \begin{cases} \frac{Q_{dep}}{(f_{wash}+f_{res})} + \left( M_{dep(t-1)} - \frac{Q_{dep}}{(f_{wash}+f_{res})} \right) \times exp(-(f_{wash} + f_{res})\delta t) \text{ if } (f_{wash} + f_{res}) \neq 0, \\ M_{dep(t-1)} + Q_{dep} \times \delta t \text{ if } (f_{wash} + f_{res}) = 0. \end{cases} \tag{8}$$

As indicated in the previous sections, $M_{dep}$ is determinant to calculate the particle resuspension and wash-off. It represents the maximal particle mass that can be resuspended, to ensure mass conservation on the street surface. Besides the respect 270 of mass conservation on the street surface, this approach presents other advantages: ($i$) particle deposition is estimated from concentrations (and not from wear emissions as in NORTRIP), following a formulation adapted to street-network geometries, and ($ii$) no additional hypothesis is necessary to determine chemical speciation of resuspended particles, as resuspended and deposited particles are assumed to have the same composition. A limitation of this approach lies in the difficulty to estimate the resuspension factor $f_{res}$ and its variations with chemical species.

To compute the concentrations in the street volume, MUNICH and SinG solves an equation describing the time evolution of the mass $M$ in each street segment (Kim et al., 2018; Lugon et al., 2020). New terms are added to this equation: the particle resuspension rate $Q_{res}$; the tyre, brake and road emission rates ($Q_{emis,wear}$) as an inlet flux, and the particle wash-off rate $Q_{wash}$ as an outlet flux. They are highlighted in Equation (9), which is solved for each size section:

$$\left. \frac{dM}{dt} \right|_{volume} = \underbrace{(Q_{inflow} + Q_{emis,exh} + \mathbf{Q_{res}} + \mathbf{Q_{emis,wear}})}_{inlet\ flux} - \underbrace{(Q_{outflow} + Q_{vert} + Q_{dep,v} + \mathbf{Q_{wash}})}_{outlet\ flux}, \tag{9}$$





with $Q_{emis,exh}$ the exhaust and non-exhaust traffic emission rates, $Q_{inflow}$ the mass inflow rate at intersections, $Q_{vert}$ the turbulent mass flux between the atmosphere and the street, $Q_{outflow}$ the outflow flux, and $Q_{dep,v}$ the deposition flux over the street volume, considering the street pavement and building walls surfaces. These terms are detailed in Kim et al. (2018) and Lugon et al. (2020). Concentrations calculated in each street using Equation (9) are then used to compute the deposition mass flux, as mentioned in Section 2.1.

## 285    3    Simulations setup

This section describes the input data and the model configuration of the SinG simulations, at both the regional and local scales. Six particle size sections are employed, with bound diameters 0.01, 0.0398, 0.1585, 0.4, 1.0, 2.5 and 10 $\mu$m. Simulations were run from 15 March to 15 May 2014. Model to data comparisons are performed from 12 April to 15 May, where BC concentrations were measured at the air monitoring station operated by Airparif during the TRAFIPOLLU project

(http://www.agence-nationale-recherche.fr/?Project= ANR-12-VBDU-0002).

### 3.1    Regional scale

SinG is employed to simulate BC concentrations over a regional-scale domain covering Paris with a spatial resolution of 1 km x 1 km (domain 4), as illustrated in Figure 1. Initial and boundary concentrations are obtained from nested simulations at the regional scale using the model Polair3D, over Europe (domain 1, spatial resolution of 55 km x 55 km), France (domain

2, spatial resolution of 11 km x 11 km), Île-de-France region (domain 3, spatial resolution of 2 km x 2 km) and Greater Paris (domain 4, spatial resolution of 1 km x 1 km). All four domains have the same vertical discretization with 14 levels, from 0 to 12 km. Meteorological data are calculated using the WRF model (version 3.9.1.1), as detailed in Lugon et al. (2020).

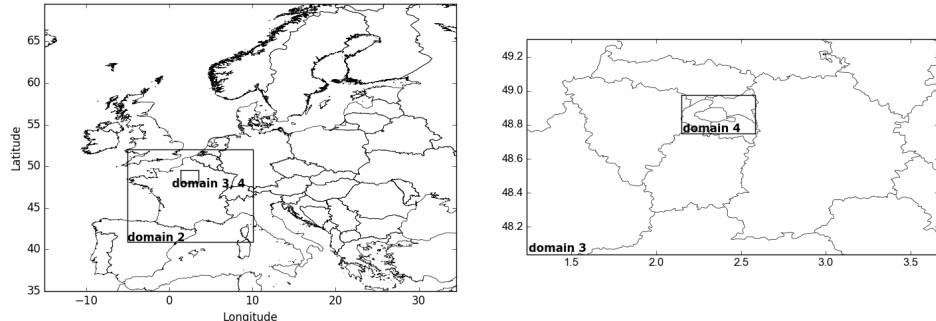

**Figure 1.** Regional-scale nested domains: Europe (domain 1, spatial resolution 55 km x 55 km), France (domain 2, spatial resolution 11 km x 11 km), Île de France region (domain 3, spatial resolution 2 km x 2 km) and Greater Paris (domain 4, spatial resolution 1 km x 1 km).

BC emissions over Europe (domain 1) and France (domain 2) are obtained from the European Monitoring and Evaluation Program (EMEP) emission inventory for the year 2014, with a spatial resolution of 0.1° x 0.1°.



Over the Île-de-France region (domain 3) and Greater Paris (domain 4), sources other than traffic are obtained using the AIRPARIF emission inventory of 2012. Because the simulated period corresponds to spring/summer in France with low contribution of the residential sector, the differences between the emissions of 2012 and 2014 for sectors other than traffic have a low impact on BC modelling over Paris. BC traffic emissions are computed using the emission inventory of 2014 provided by the air-quality agency of Paris (AIRPARIF), except in the street-network region (see Figure 2), where they are obtained from

the TRAFIPOLLU project, as detailed in the section "Local scale" below. For BC traffic non-exhaust emissions, over domain 4 (both in the street-network region and outside), different scenarios of non-exhaust emissions are studied, as described in the section 4 (see Table 1).

     Note that in SinG, which is used over domain 4, traffic emissions are not used for the regional-scale modelling in the street-network region, but only for the local-scale (street) modelling, because the local and regional scales are two-way coupled.

**3.2   Local scale**

At the local scale, SinG simulates the street network represented in Figure 2, containing 577 streets including the "Boulevard Alsace Lorraine", where the measurements were performed. Meteorological data above each street are extracted from the same WRF simulation as employed at the regional scale.

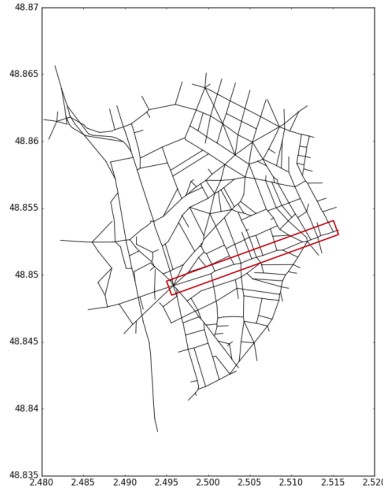

**Figure 2.** The street network, with the "Boulevard Alsace Lorraine" highlighted in the red rectangle.

     BC exhaust traffic emissions are obtained from the TRAFIPOLLU project, using the same street network and exhaust

inventory as detailed in Kim et al. (2018): the dynamic traffic model SymuVia was used to determine the hourly traffic in each street of the network and pollutant emissions were deduced using COPERT IV emissions factors; emissions were calculated for two typical days: 25 March 2014, representative of weekdays, and 30 March 2014, representative of a week-ends and holidays. BC non-exhaust emission factors from tyre, brake and road wear are computed using the formulations proposed in the EMEP





guidelines (Ntziachristos and Boulter, 2016) to calculate $PM_{10}$ wear emissions, and a speciation represented by a BC fraction

($f_s^{bc}$), defined for each wear source. The values used in this study for $EF_{s,v}^{tsp}$, $f_s^{pm_{10}}$ and $f_s^{bc}$ are detailed in section 4.

$$EF_{s,v}^{bc} = EF_{s,v}^{tsp} \times f_s^{pm_{10}} \times S_s(u_v) \times f_s^{bc}. \tag{10}$$

where $s$ is the non-exhaust source (tyre, brake or road wear, indicated by $ty$, $bk$ and $rd$ respectively), $v$ is the vehicle type ($v$ = LDV or HDV), $EF_{s,v}^{tsp}$ is the total suspended particles (TSP) wear emission rate for a determined source $s$ and vehicle $v$, $f_s^{pm_{10}}$ the $PM_{10}$ fraction in each wear source $s$, and $S_s$ the correction factor according to vehicle speed $u_v$. $S_s$ varies according

to the wear emission source, as shown in Equations (11), (12) and (13):

$$S_{ty}(u_v) = \begin{cases} 1.39 \text{ if } u_v < 40\text{km.h}^{-1}, \\ -0.00974u_v + 1.78 \text{ if } 40\text{km.h}^{-1} \le u_v \le 90\text{km.h}^{-1}, \\ 0.902 \text{ if } u_v > 90\text{km.h}^{-1}, \end{cases} \tag{11}$$

$$S_{bk}(u_v) = \begin{cases} 1.67 \text{ if } u_v < 40\text{km.h}^{-1}, \\ -0.0270u_v + 2.75 \text{ if } 40\text{km.h}^{-1} \le u_v \le 95\text{km.h}^{-1}, \\ 0.185 \text{ if } u_v > 95\text{km.h}^{-1}, \end{cases} \tag{12}$$

$$S_{rd}(u_v) = 1.0. \tag{13}$$

The total suspended particles wear rates emitted by tyre and brake wear in HDV ($EF_{s,hdv}^{tsp}$) are computed as a function of

passenger cars (PC) wear emissions. They take into account the vehicle characteristics as HDV weight ($LCF_s$) and the number of axles ($N_{axle}$), as indicated in Equations (14) and (15).

$$EF_{ty,hdv}^{tsp} = \frac{N_{axle}}{2} \times LCF_s \times EF_{ty,pc}^{tsp}, \tag{14}$$

$$EF_{bk,hdv}^{tsp} = 3.13 LCF_s \times EF_{bk,pc}^{tsp}, \tag{15}$$

with $LCF_s$ a load correction computed as a function of the load factor $LF$, ranging from 0 and 1.

$$LCF_s = \begin{cases} 1.41 + (1.38LF) \text{ for tyre wear }, \\ 1.00 + (0.79LF) \text{ for brake wear }. \end{cases} \tag{16}$$





The resulting non-exhaust emission rate ($Q_{emis,wear}$) for all wear emission source $s$ is proportional to the LDV and HDV vehicle flow ($N_v$), as indicated in Equation (17):

$$Q_{emis,wear} = \sum_{s=1}^{3} \sum_{v=1}^{2} \left( EF_{s,v}^{bc} \times N_v \right). \tag{17}$$

As particle resuspension is directly dependent on the particle mass deposited on the street surface, it is dependent on the
initial conditions of the simulation. In order to avoid this dependence, a spin up of 25 days is adopted to enable a mass surface equilibrium between deposition, wash-off and resuspension mass rates. More details are provided in section 4.3.

## 4   Sensitivity analysis to black carbon non-exhaust emissions

As mentioned in the introduction, non-exhaust emissions are difficult to estimate and control. The huge variability of car models and speed regimes, combined to a large diversity of tyre, brake and road components contribute to the large uncertainties
associated to non-exhaust emissions. Furthermore, experimental studies report the complexity to differentiate resuspension from tyre and road-wear emissions.

This section presents a sensitivity analysis of BC street concentrations to BC non-exhaust emissions, using different tyre, brake and road emission factors from the literature.

### 4.1   The simulations

Different simulations were performed, with large variations in tyre-wear emissions properties. The BC fraction adopted in brake and road wear are constant in all simulations, and follow the EMEP guidelines (Ntziachristos and Boulter, 2016) with $f_{bk}^{bc} = 0.026$ and $f_{rd}^{bc} = 0.0106$. Table 1 summarises the configuration used in each simulation, with the $PM_{10}$ fraction employed in each source $s$ ($f_s^{pm_{10}}$), the BC fraction adopted in tyre-wear emissions ($f_{ty}^{bc}$), and the resultant BC emission factor from each vehicle type $v$ (LDV and HDV) and non-exhaust source $s$ ($EF_{v,s}^{bc}$).

To show the influence of non-exhaust emissions, simulation 1 ignores non-exhaust emissions. Simulations 2 employs the BC wear emission factors indicated in the EMEP guidelines, also used in the HERMES model (Guevara et al., 2019; Ntziachristos and Boulter, 2016). TSP tyre wear emission factors from HDV vehicles $EF_{hdv,ty}^{tsp}$ are deduced from Equations (14), considering a load factor $LF$ equals to 1.0, and a number of axles $N_{axle}$ equals to 2.0. Simulation 3 uses the $PM_{10}$ wear emission factors indicated in the NORTRIP model (Denby et al., 2013a). They are of the same order of magnitude than those of the EMEP
guidelines. Note that for road-wear emissions, the soil characteristics used in Bouvelard Alsace Lorraine by Thouron et al. (2018) were employed, leading to higher road-wear emissions than in the EMEP guidelines. As NORTRIP does not indicate the BC fractions from each source, the same BC fractions as in the EMEP guidelines (Ntziachristos and Boulter, 2016) are adopted. Simulation 4 employs the same brake and road-wear emission factors as in the EMEP guidelines, but tyre-wear emission factors are higher. A LDV tyre emission factor $EF_{ldv,ty}^{tsp}$ of 100 mg.vkm$^{-1}$ is used, as proposed in several studies
Malmqvist (1983); Baumann and Ismeier (1997); Gebbe (1997); Luhana et al. (2004); Boulter (2005). As in simulation 2, the TSP tyre-wear emission factors from HDV vehicles $EF_{hdv,ty}^{tsp}$ are deduced from Equations (14), considering a load factor





$LF$ equals to 1.0, and a number of axles $N_{axle}$ equals to 2.0. A $PM_{10}$ fraction $f_{ty}^{pm_{10}}$ 0.6 is adopted, as in simulation 2. The tyre-wear BC fraction $f_{ty}^{bc}$ is higher than in simulations 2 and 3. It is taken equal to 0.25, which is the average of the BC mass weight fraction in tyre compositions (between 22%-30%) indicated in Thorpe and Harrison (2008). As in the EMEP guidelines

and simulation 2, a speed correction factor $S_{ty}(u_v)$ of 1.39 is employed in the simulation, as the LDV and HDV vehicle speed in the street network are lower than 40 km.h$^{-1}$ (average speed around 32 km.h$^{-1}$ according to TRAFIPOLLU measurements). The resulting $PM_{10}$ and BC tyre-wear emission factors from LDV and HDV are then $EF_{ldv,ty}^{pm_{10}} = 83.4$ mg.vkm$^{-1}$, $EF_{hdv,ty}^{pm_{10}} = 232.7$ mg.vkm$^{-1}$, $EF_{ldv,ty}^{bc} = 20.8$ mg.vkm$^{-1}$ and $EF_{hdv,ty}^{bc} = 57.1$ mg.vkm$^{-1}$.

In order to evaluate the impact of particle resuspension on BC concentrations in streets, simulation 5 is similar to simulation

4, but it does not take into account particle deposition. In other words, simulation 5 simulates a maximal resuspension rate, equals to the deposition rate. Finally, in order to evaluate the influence of uncertainties in the BC speciation of exhaust emissions, simulation 6 uses the same non-exhaust emission factors as simulation 4, but BC exhaust emission factors are artificially increased by 23%. This correction factor is defined based on the traffic-flow characteristics observed in the "Boulevard Alsace Lorraine" during the TRAFIPOLLU campaign, and the BC/$PM_{2.5}$ uncertainties for each vehicle class detailed in Table 3-91

of Ntziachristos and Samaras (2018).

**Table 1.** List of the simulations performed, with the configuration options (with or without deposition, with or without exhaust emission correction), the $PM_{10}$ fraction ($f_s^{pm10}$) adopted in each wear emission source (tyre wear indicated by $ty$, brake wear indicated by $bk$, and road wear indicated by $rd$), the BC fraction adopted in tyre-wear emissions ($f_{ty}^{bc}$), and the correspondent BC emission factors for each wear-emission source and vehicle type $hdv$ and $ldv$ ($EF_{v,s}^{bc}$) in mg.vkm$^{-1}$.

| Sim. | with dep. | with exh. emis. cor. | $f_{ty}^{pm_{10}}$ | $f_{ty}^{bc}$ | $EF_{ldv,ty}^{bc}$ | $EF_{hdv,ty}^{bc}$ | $f_{bk}^{pm_{10}}$ | $EF_{ldv,bk}^{bc}$ | $EF_{hdv,bk}^{bc}$ | $f_{rd}^{pm_{10}}$ | $EF_{ldv,rd}^{bc}$ | $EF_{hdv,rd}^{bc}$ |
|------|-----------|---------------------|--------|--------|---------|---------|--------|---------|---------|--------|---------|---------|
| 1 | yes | no | 0.00 | 0.00 | 0.00 | 0.00 | 0.00 | 0.00 | 0.00 | 0.00 | 0.00 | 0.00 |
| 2 | yes | no | 0.60 | 0.153 | 1.36 | 3.81 | 0.98 | 0.32 | 1.79 | 0.50 | 0.08 | 0.40 |
| 3 | yes | no | 0.10 | 0.153 | 0.71 | 3.58 | 0.80 | 0.21 | 1.04 | 0.18 | 0.47 | 2.40 |
| 4 | yes | no | 0.60 | 0.250 | 20.8 | 57.1 | 0.98 | 0.32 | 1.79 | 0.50 | 0.08 | 0.40 |
| 5 | no | no | 0.60 | 0.250 | 20.8 | 57.1 | 0.98 | 0.32 | 1.79 | 0.50 | 0.08 | 0.40 |
| 6 | yes | yes | 0.60 | 0.250 | 20.8 | 57.1 | 0.98 | 0.32 | 1.79 | 0.50 | 0.08 | 0.40 |

## 4.2   Model to measurement comparisons

The different simulations are evaluated by model to measurement comparisons of BC concentrations, measured at "Boulevard Alsace Lorraine". The statistical criteria applied to evaluate the model performance are those defined by Hanna and Chang (2012) and Herring and Huq (2018). Two different criteria are defined, a most strict criteria, with $-0.3 <$ FB $< 0.3$; $0.7 <$ MG





< 1.3; NMSE < 3; VG < 1.6; FAC2 ≥ 0.5; NAD < 0.3, and a less strict criteria acceptable in urban areas, with -0.67 < FB < 0.67; NMSE < 6; FAC2 ≥ 0.3; NAD < 0.5 [1]. The definitions of the statistical indicators are given in Annex A1.

Table 2 shows the statistical indicators obtained from the model to measurement comparisons for each simulation of Table 1. The BC concentrations observed at "Boulevard Alsace Lorraine" are strongly underestimated in simulations 1, 2 and 3, with a fractional bias (FB) equal to -1.26, -1.10 and -1.15 respectively, not satisfying any performance criterion. The configuration
used in simulation 4, with higher tyre and brake-wear emissions, results in satisfactory statistical indicators, respecting all the most strict performance criteria proposed by Hanna and Chang (2012) and Herring and Huq (2018). The temporal evolution of BC concentrations obtained with simulation 4 is illustrated in Figure 3, showing a good correlation between the BC hourly concentrations observed and those simulated by SinG. Both the NORTRIP and the HERMES models, whose emission factors are used in simulations 1, 2 and 3, have quite low $PM_{10}$ tyre-wear emission factors compared to the simulations 4, 5 and 6,
which use higher tyre-wear emission factor for passenger cars, as proposed in the review Boulter (2005). Previous simulations with the NORTRIP and HERMES models (Denby et al., 2013a; Pay et al., 2011) achieved good correlations between simulated and measured particle concentrations, because they assume that resuspension is the main non-exhaust emission process and use high resuspension rates. The $PM_{10}$ resuspension rates used by Pay et al. (2011) (88 mg.vkm$^{-1}$ for LDV and 217 mg.vkm$^{-1}$ for HDV) are similar to the tyre-wear emission factors employed in simulations 4, 5 and 6 (83.4 mg.vkm$^{-1}$ for LDV and 232.7
mg.vkm$^{-1}$ for HDV). These high resuspension rates may compensate the low non-exhaust tyre-wear emissions employed in the HERMES and NORTRIP models. The ratio between the total BC emissions in simulation 4 and simulation 2 is 3.30. A quite similar ratio of 3.0 was employed by Brasseur et al. (2015) to increase BC emissions from COPERT-IV emission inventory, as mentioned in the introduction.

**Table 2.** Comparisons to BC measurements at "Boulevard Alsace-Lorraine": statistical indicators obtained for the different SinG simulations.

|   | o [$\mu$g.m$^{-3}$] | s [$\mu$g.m$^{-3}$] | FB | MG | NMSE | VG | FAC2 | NAD |
|---|---|---|---|---|---|---|---|---|
| 1 | 6.07 | 1.37 | -1.26 | 0.22 | 3.69 | 11.62 | 0.04 | 0.63 |
| 2 | 6.07 | 1.74 | -1.10 | 0.29 | 2.54 | 5.80 | 0.12 | 0.55 |
| 3 | 6.07 | 1.62 | -1.15 | 0.27 | 2.85 | 6.97 | 0.09 | 0.58 |
| 4 | 6.07 | 4.91 | -0.21 | 0.82 | 0.29 | 1.27 | 0.77 | 0.19 |
| 5 | 6.07 | 4.92 | -0.20 | 0.82 | 0.29 | 1.27 | 0.77 | 0.19 |
| 6 | 6.07 | 5.12 | -0.16 | 0.85 | 0.27 | 1.26 | 0.78 | 0.18 |

The comparison between simulations 4 and 5 shows that particle dry-deposition has a low impact on BC concentrations
in the streets (0.15% on average). Because the BC mass that may be resuspended is limited by the BC deposited mass, this means that BC resuspension is also not significant compared to the other BC emission sources. Figure 5 illustrates the temporal evolution of BC deposition, wash-off and resuspension. The resuspension rate is limited by the deposited mass at the surface, and both resuspension and deposition rates are of the same order of magnitude. BC wash-off mass rates present the same order

[1]FB represents the fractional bias, MG the geometric mean bias, NMSE the normalised mean square error, VG the geometric variance, NAD the normalised absolute difference, and FAC2 the fraction of predictions within a factor two of observations.



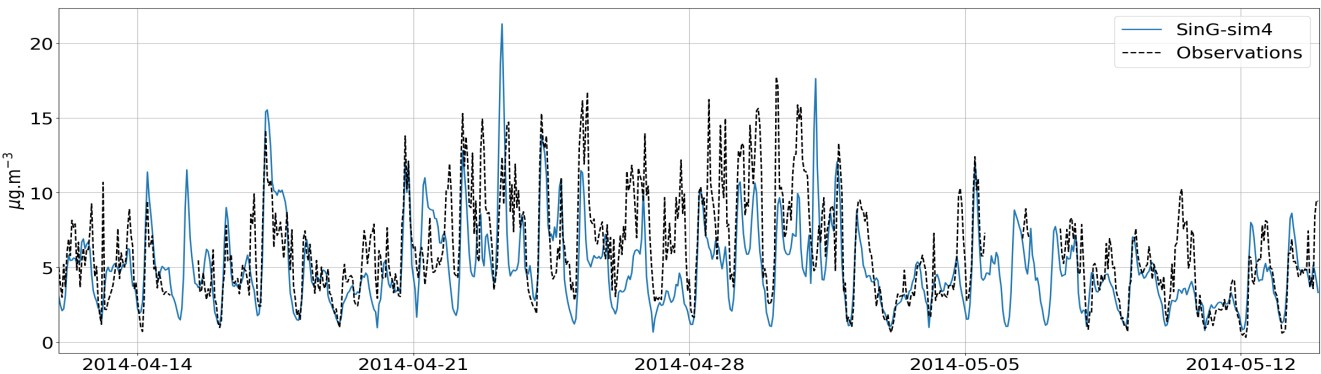

**Figure 3.** BC hourly concentrations at "Boulevard Alsace Lorraine" simulated by SinG (simulation 4) and observed in the TRAFIPOLLU campaign [$\mu$g.m$^{-3}$].

410 of magnitude of reposition and resuspension processes, and are concentrated in raining days. This finding of low resuspension rate agrees with the observation of Luhana et al. (2004), which estimate that tyre-wear emissions are the most important non-exhaust emissions, and that particle resuspension is not very important.

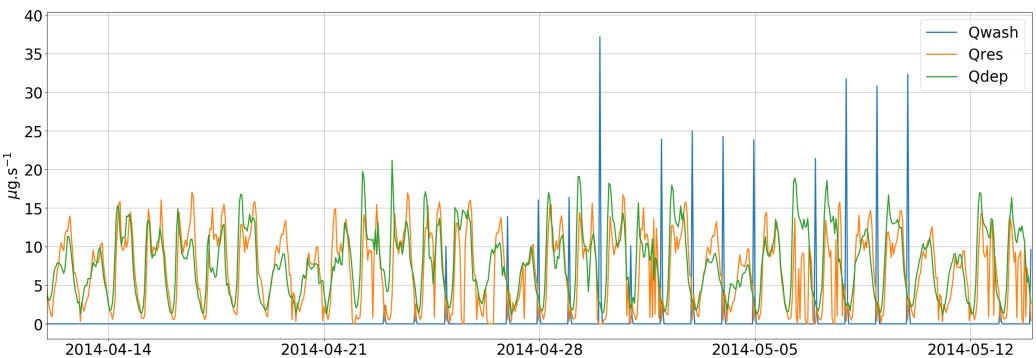

**Figure 4.** BC wash-off, deposition and resuspension rates simulated by SinG (simulation 4) [$\mu$g.s$^{-1}$].

 The comparison between simulations 4 and 6 shows that increasing BC exhaust emissions by taking into account the uncertainties of the BC speciation in exhaust emissions lightly improves the BC concentrations in streets. Differences in BC concentrations simulated by modifying the BC exhaust emissions are less important than the differences simulated by modify-
415 ing the BC tyre-wear emissions, suggesting that non-exhaust emissions are the most crucial emissions to improve.

### 4.3 Deposited mass over streets

A good representation of the particle deposited mass over the street surface is determinant to calculate the particle resuspension. This section compares the simulated particle deposited mass over the " Alsace Lorraine" to the one observed by Hong et al.





(2016a) during the TRAFIPOLLU campaign. Hong et al. (2016a) collected *in-situ* the total road dust with dry sampling by a

vacuum cleaner on 14 October 2014 at different points of the avenue. Figure 8a of Hong et al. (2016a) indicates the estimated total weight of dry stocks (in kg), ranging from 1 to 21 kg and with an average value about 8.90 kg. Considering the road surface is 2661 m$^2$, as indicated in Hong et al. (2016a), the TSP mass density in "Boulevard Alsace Lorraine" is 3.34 g.m$^{-2}$. The size distribution of these samples is presented in Figure 5 of Hong et al. (2016b), indicating that approximately 7.5% of the total deposited mass is composed of particles of diameters lower than 10 $\mu$m (PM$_{10}$). Therefore, the average mass density

of fine particles (PM$_{10}$) in "Boulevard Alsace Lorraine" is about 250 mg.m$^{-2}$. Note that quite similar PM$_{10}$ mass density was observed by Amato et al. (2009) in Barcelona urban areas (ranging from 8.9 to 216 mg.m$^{-2}$). In the simulations, to deduce the PM$_{10}$ deposited mass from the BC deposited mass, the fraction of BC in the PM$_{10}$ deposited mass at the surface is assumed to be 2.11%, following the observations of the chemical patterns of PM$_{10}$ in deposited road dust in urban environment performed by Amato et al. (2009).

Figure 5 illustrates the PM$_{10}$ deposited mass density (in mg.m$^{-2}$) obtained with each simulation at "Boulevard Alsace Lorraine". As mentioned in section 3.2, a spin-up of 25 days is used in each simulation, so that the deposited mass is independent of the initial conditions. Hence, here model to measurement comparisons are performed starting from 12 April. In Figure 5, this initial time of model to measurement comparisons (12 April) is indicated by the black vertical line. This spin-up is important to avoid an underestimation of particle deposition due to initial conditions. Table 3 summarizes the average PM$_{10}$ mass density

obtained from 12 April to 14 May in all simulations, the PM$_{10}$ mass density estimated using the measurements performed by Hong et al. (2016a, b), and the corresponding FB between these mass densities. A good model to data correspondence is observed for simulations 4 and 6 with FB = -0.16 and -0.06 respectively. In simulations 1, 2 and 3, the particle deposited mass at the street surface is strongly underestimated, with a FB around -0.60. Note that simulation 5 does not consider particle deposition, so no particle deposited mass is present at the street surface.

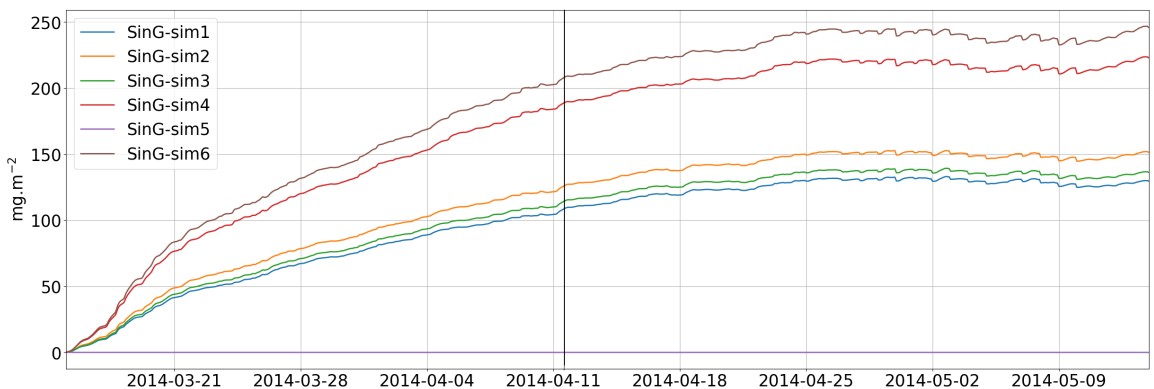

**Figure 5.** Temporal evolution of equivalent PM$_{10}$ deposited mass density over the "Boulevard Alsace Lorraine" surface obtained with the different SinG simulations (in mg.m$^{-2}$).





**Table 3.** Average mass density of $PM_{10}$ deduced from the measurements at "Boulevard Alsace Lorraine" (obs. $PM_{10}$) [mg.m$^{-2}$], average simulated mass density of $PM_{10}$ (sim $PM_{10}$) [mg.m$^{-2}$], and the fractional bias (FB) [.].

|       | obs $PM_{10}$ | sim $PM_{10}$ | FB    |
|-------|---------------|---------------|-------|
| sim-1 | 250           | 125.9         | -0.66 |
| sim-2 | 250           | 145.3         | -0.53 |
| sim-3 | 250           | 132.0         | -0.62 |
| sim-4 | 250           | 212.3         | -0.16 |
| sim-5 | 250           | 0.0           | -2.00 |
| sim-6 | 250           | 234.1         | -0.06 |

## 5 Influence of the two-way coupling on BC concentrations in streets

This section investigates the influence of the two-way coupling between the regional and local scales on BC concentrations in the street network by comparing the concentrations simulated by SinG and MUNICH. As mentioned in section 2, the coupling between the local and regional scales is two way in SinG, which couple the street model MUNICH to the chemistry transport model Polair3D. However, MUNICH may be used as a standalone model, simulating the street concentrations with a one-way coupling to the regional-scale background concentrations. In that case, the regional-scale (background) concentrations influence the street concentrations, but the street concentrations do not influence the background concentrations, and the vertical mass transfer between local and regional scales only influences concentrations in the streets. Note that in the one-way coupling, traffic emissions are used both in the regional-scale model Polair3D and in the street model MUNICH. Because SinG employs a two-way coupling, at each time step the vertical mass transfer between local and regional scales enables to calculate concentrations in the background and in streets, providing a direct interaction between concentrations in the street network and those in the urban background. Therefore, traffic emissions are considered only at the local scale, and there is no double counting of traffic emissions in SinG.

Simulations of BC concentrations using Polair3D, MUNICH and SinG are performed using the non-exhaust emission factors of simulation 4 (see Table 1). As in Lugon et al. (2020), two factors are analyzed to investigate the influence of the two-way coupling on the local-scale concentrations: ($i$) the strength of traffic emissions, and ($ii$) the street aspect ratio $\alpha_r$ (ratio between street height and width). The vertical mass flux between the local and regional scales is proportional to the concentration gradient between the street and the urban background, as shown in Equation (8) of Lugon et al. (2020). Streets with high traffic emissions tend to favor the vertical mass transfer from the local to the regional scales, as they tend to present a high gradient between the street and the urban background concentrations. This vertical mass flux is also dependent of the street geometry, represented by the aspect ratio $\alpha_r$ (see Equation 9 in Lugon et al. (2020)). Streets with low aspect ratio (large streets) tend to favor the vertical transfer between local and regional scales, and in streets with high aspect ratios (narrow streets) the vertical mass transfer between scales tends to be lower. Streets are classified as large streets when $\alpha_r \leq 0.3$, intermediate streets when $0.3 < \alpha_r < 0.6$, and narrow streets when $\alpha_r \geq 0.6$ (Kim et al., 2018).





Figures 6 illustrates, respectively, the average BC emissions in the street network (panel (a)), the average BC concentrations simulated by SinG (panel (b)), the absolute difference between the BC concentrations simulated by SinG and MUNICH (panel (c)), and the absolute relative differences between the BC concentrations simulated by SinG and MUNICH (panel (d)). A good correlation between traffic emissions and concentrations is observed, as expected. The differences between the average BC concentrations simulated by SinG and MUNICH range from -2.6 $\mu$g.m$^{-3}$ to 2.4 $\mu$g.m$^{-3}$, corresponding to relative errors between 0% and 50%. The BC concentrations simulated by SinG are higher than those simulated by MUNICH especially in the streets located in the left side of the street network, where the traffic emissions are the highest. This effect is also observed in the streets adjacent to the high-emission streets, probably due to transport of pollutants from one street to another. Even though the BC concentrations are higher in SinG than in MUNICH near streets with high traffic emissions, they are lower in a large part of the domain, in the middle of the street network. This region presents simultaneously low BC emissions and concentrations, and high aspect ratios $\alpha_r$. Therefore, the concentration gradients between the local and regional scales are low, leading to low differences between local and regional-scale concentrations. However, the absolute values of the relative differences between the concentrations simulated by MUNICH and SinG in this region are large and reach 50%. These large differences and the lower concentrations simulated in MUNICH compared to SinG may be explained by the double counting of traffic emissions performed by MUNICH with the one-way coupling technique.

In order to evaluate more precisely the influence of the street geometry (represented by $\alpha_r$) in the relative differences between the BC concentrations simulated by SinG and MUNICH, the analysis of different $\alpha_r$ intervals is made. Figure 7 illustrates the percentage of streets present in different ranges of $\alpha_r$ intervals encountered in the street network for different ranges of relative differences between the BC concentrations simulated by SinG and MUNICH. The sum of each column corresponds to 100%. The streets with low values of $\alpha_r$ (large streets) tend to have more cars passing and higher emission rates. In this case, the BC concentrations tend to be higher using SinG than using MUNICH, and the relative differences are mostly positives. The higher the street aspect ratio $\alpha_r$ is, the higher the percentages of streets with negative relative differences between SinG and MUNICH are. The impact of the two-way coupling on BC concentrations in streets is high, with relative differences ranging from -51% to 48%.

## 6 Conclusion

This study performs simulations of BC concentrations in a Parisian suburb street-network using the multi-scale model SinG. Two main aspects influencing the BC concentrations in streets are investigated: the non-exhaust emissions (tyre, brake and road wear) and particle resuspension, and the two-way coupling between the street-network model and chemistry transport model that computes the urban background concentrations. A new parametrization to calculate particle resuspension is presented, respecting the mass balance at the street surface.

Constant resuspension emission factors are often used in the literature to model resuspension, without taking into account the amount of deposited mass. Particle resuspension is not an important source of BC in the simulations here, because the deposited mass over the street surface is not significant enough to justify high constant resuspension emission factors. As





**Figure 6.** Average BC emissions in the street-network (a), average BC concentrations simulated by SinG (b), absolute differences between the BC concentrations simulated by SinG and MUNICH (c), and absolute values of relative differences between the BC concentrations simulated by SinG and MUNICH (d).

the simulated deposited mass agrees well with in-situ measurements, other sources of BC emissions are investigated here. In particular, non-exhaust emissions, such as tyre wear, can be relevant to improve the modelling of BC concentrations in streets. They can be as relevant as exhaust emissions, and their underestimation may justify the virtual increase of BC emissions often employed in street-network modelling studies. Non-exhaust emissions still present very high uncertainties, with a large spectrum of emission factors and experimental limitations to separate each non-exhaust emission source. More studies are needed to better characterize these emissions, regarding their size distributions and chemical compositions. In particular, large differences are observed in the literature, between different studies estimating tyre-wear emissions, with tyre-wear emission factors from the EMEP guidelines being on the low side of factors. Following the literature, increasing the BC passenger cars tyre-wear emission factors of the EMEP guidelines from 1.36 mg.vkm$^{-1}$ to 20.8 mg.vkm$^{-1}$ lead to good comparisons of the simulated BC concentrations to the measured ones.





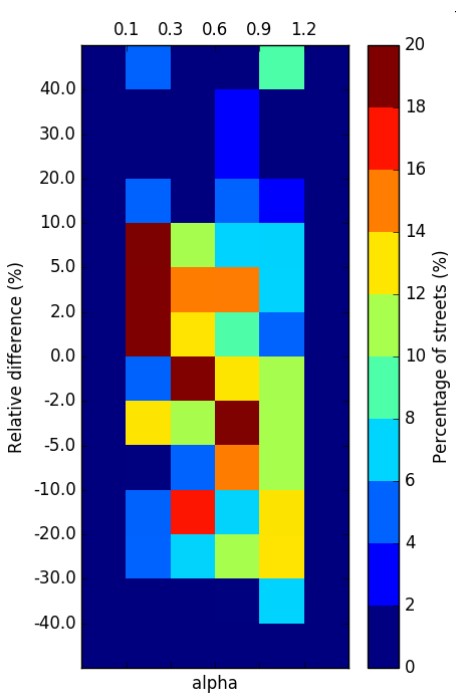

**Figure 7.** Percentage of streets (purple) present in different $\alpha_r$ intervals for different ranges of the relative differences between the BC concentrations simulated by SinG and MUNICH.

The two-way coupling rather than one-way coupling between the local and regional scales proved to strongly influence the BC concentrations in streets, because the gradients of concentrations between the local-scale street concentrations and the regional-scale background concentrations are often high. The intensity of the vertical mass flux between streets and background are function of traffic emissions and street geometry (represented by the aspect ratio $\alpha_r$). The two-way coupling leads to higher BC concentrations in large streets with high traffic emissions, and lower BC concentrations in narrow streets with low traffic emissions ( with differences ranging from -51.5% to 48.4%).

## Appendix A: Statistical parameters

### A1 Definitions

– FB: Fractional bias

$$FB = 2\left(\frac{\bar{o} - \bar{c}}{\bar{o} + \bar{c}}\right)$$

– MG: Geometric mean bias

$$MG = exp(\overline{ln(o)} - \overline{ln(c)})$$





- NMSE: Normalized mean square error

520 $$NMSE = \frac{\overline{(o-c)^2}}{\overline{oc}}$$

- VG: Geometric variance

$$VG = exp[\overline{(ln(o)-ln(c))^2}]$$

- NAD: Normalised absolute difference

$$NAD = \frac{\overline{|c-o|}}{\overline{(c+o)}}$$

525 - FAC2: Fraction of data that satisfy

$$0.5 \leq \frac{c}{o} \leq 2.0$$

Where $o$ and $c$ represent the observed and simulated concentrations respectively.

*Author contributions.* KS and LL were responsible for conceptualization. LL developed the software. LL conduced the visualization and validation; LL and KS performed the formal analysis. JV, KS and CD acquired resources. KS and OC were responsible for funding acquisition.
530 LL and KS were responsible for writing and original draft preparation.

*Competing interests.* The authors declare that they have no conflict of interest.

*Data availability.* The source code of Street-in-Grid and MUNICH models are available via Zenodo with the following DOI https://doi.org/10.5281/zenodo.4393638. The observation data are available from AIRPARIF, Air quality – TRAFIPOLLU campaing. They are available upon request.

535 *Acknowledgements.* This research has been supported by the Department of Green Spaces and Environment (Mairie de Paris) and the École des Ponts ParisTech (grant CIFRE no. 2017/064). The authors thank Airparif for providing the emission inventory and the measured concentration data, and the TRAFIPOLLU ANR project for making data available for the model application and evaluation.



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
