# Peer review of "Black carbon modelling in urban areas: investigating the influence of resuspension and non-exhaust emissions in streets using the Street-in-Grid model for inert particles (SinG-inert)"

_Geoscientific Model Development, 2020_

## Short Comment (SC1) · 26 Feb 2021

Dear authors,

in my role as Executive editor of GMD, I would like to bring to your attention our Editorial version 1.2:

https://www.geosci-model-dev.net/12/2215/2019/

This highlights some requirements of papers published in GMD, which is also available

on the GMD website in the 'Manuscript Types' section:

http://www.geoscientific-model-development.net/submission/manuscript_types.html

In particular, please note that for your paper, the following requirement has not been met in the Discussions paper:

- "The main paper must give the model name and version number (or other unique identifier) in the title."

Please add a version number for SinG in the title upon your revised submission to GMD.

Yours,

Astrid Kerkweg
* * *

---

## Referee Comment (RC1) · Anonymous Referee #1 · 16 May 2021

The underestimation of BC concentration in air quality models has been a comment issue and this work provides some perspectives that would be important to solve this problem. Those founding are important and could be of the interest of readers and scientific communities in this field. However, the languages and overall structure of the manuscript should be improved before publications. My comments are as follows:

Major comments:

1. The introduction is too long and requires serious streamlining. I would suggest

the author summarize those modeling literature and emission numbers using one or two tables. And only discuss the technical difference between those works. Also, to make the overall structure more clear, the authors could consider rearranging the literature part based on the following order: observations, modeling methods, model performance, gaps need to be addressed in current models.

2. The description of different simulations conducted in this study is confusing. After careful and difficult reading, I guess there are two different sets of simulations. One set is 6 simulations with Polairs3D-SinG-MUNICH model, another one is 1 simulation with Polairs3D-MUNICH model. If that is the case, the author should make it more clear in Section 2. Also, the author should improve the simulation scenario description by adding key massages in Table 1, such as the model/source used for wear emission calculation.

3. The author should explain why the simulation results present in Section 5 are not evaluate with observations. Without the model performance compared, it is impossible to evaluate the necessity or advantage to adopt the two-way coupling method compares to the one-way coupling method.

4. The author takes the entire section 2 to describe a new "deposition-resuspension" framework but concludes that the whole "deposition-resuspension" process is insignificant in improving BC underestimation (line 405-410). Instead, the selection of emission factor is the key to solve the problem. This made the work of the new "deposition-resuspension" framework meanless. The author could add one more simulation with the setting of simulation 4 while turn off the new "mass-conserved deposition-resuspension" framework, to properly evaluate the potential of this framework. If not, I would suggest the author move entire section 2 to the appendix and only briefly introduce the new framework in the "Simulation setup" section.

5. In general, the languages used in this manuscript could be improved. Try to avoid long sentences with multiple subordinate clauses that really made the text difficult to

follow.

Minor comments:

1. Line 7-9, the statement after "i.e." is confusing, please rephrase. Do you mean the BC from one street can be transported to another street at regional scales?

2. Line 22-23: Please rephrase. For example, Here comparisons are performed using ... factors from literature, and we found ?? are improved?

3. Line 58: Can you provide the LDV and HDV definition (citation?), why the MDV (mid-duty vehicles) is not considered in this study?

4. Line 75-80: Feels like most of the non-exhaust important studies are PM10 focused. Can you discuss the size distribution difference between exhaust and non-exhaust emissions?

5. Line 84-85: Is it the same ratio for both PM2.5 and PM10.

6. Line 87: How is this related to non-exhaust emission? What is the source of this species?

7. Line 91: "Some models" Please provide citations. And also, I don't think even if is appropriate here, most would be a better wording.

8. Line 92: "HERMES" Please provide a citation of this model here.

9. Line 103: Why is it supposed to be independent? Are describing the NORTRIP assumption or making general claims? Please clear, if it is the latter, you would need a citation to justify your claims.

10. Line 317: Why selecting those two days? Are they more representative of the general average?

11. Line 324: "PM10 fraction" fraction of what? PM10 fraction of TSP?

12. Line 330: What is LCFs stand for by each letter?

13. Section 4.1: Not clear if those 6 simulations are conducted based on Polairs3D-MUNICH-SinG model combination or on a simple SinG local/box model.

14. Line 350: How dose those fractions applied for different size bins?

15. Table 1: Can you also add the model SOURCE in the table? Like EMEP, HERMES, and NORTRIP?

16: Line 395-398: I don't understand here. If the non-exhaust wear emissions are very low, there should be a very small amount of mass available for resuspension. So, no matter how high the resuspension rate is, there should have no mass supply for resuspension, as demonstrated by simulation 5. So, how could a high resuspension rate along explained the much higher BC concentration in previous simulations? Isn't it just contradict your simulation 5?

17. Line 442: Very confusing statement here, do you mean SinG and MUNICH are two different systems that can replace each other? But based on section 2, MUNICH is part of SinG. Or, maybe you want to say, the comparison is between Polair3D-SinG-MUNICH and Polair3D-MUNICH (without SinG)? Same question for line 465.

18. Line 444-448: What is the point to mention the one-way feedback here? Is it used in this study? Do you mean the Polair3D-SinG-MUNICH is using two-way feedback, and Polair3D-MUNICH is using one-way feedback? And you are comparing between those two methods? 19: Line 469-470: Can you explain why this SinG results to higher BC concentrations compares to MUNICH for those streets?

20: Line 477-478: How do this double-counting works? Please provide more detailed discussion here. Why does this not seeing on high emission street?

21. Section 5: Is the SinG simulation in Section 5 the same as simulation 4 in section 4? Is the MUNICH simulation in section 5 have the same model setup for all other SinG simulations? Why simulation result from the MUNICH simulation not compared with street-level observation (Boulevard Alsace Lorraine) as in Section 4? There is no

way to judge which method has better model performance MUNICH or SinG.

---

## Referee Comment (RC2) · Anonymous Referee #2 · 30 Jul 2021

General Comments

This research article provides new and very useful information regarding the contribution of on-road traffic to urban air pollution. The focus is on particulate black carbon (BC) and its vehicular emission sources. The authors use a novel advanced modeling technique, the SinG model, which simulates urban air pollution at various spatial scales (urban background and street-level pollution) in an internally consistent manner. Furthermore, they improved this model by implementing a more detailed dry deposition
model and conducting a dynamic simulation of the emission, deposition, and resuspension of particulate matter.

In this work, they apply this model to investigate the vehicular emissions of BC. In particular, they evaluate the relative contributions of exhaust and non-exhaust (tire, brake, and road wear) emissions of BC to air pollution in streets. The non-exhaust emission source is particularly uncertain and the authors do an excellent job at reviewing the literature, identifying the major uncertainty sources, and systematically investigating their impact on simulated BC concentrations. They conduct a thorough evaluation of the modeling results using experimental results of a field program that provides both ambient air BC concentrations in a Paris suburban street and deposited PM mass in that same street. This evaluation allows the authors to propose that the greater estimates of non-exhaust emissions are more realistic and that current non-exhaust BC emission inventories may significantly underestimate actual BC vehicular emissions. This conclusion is consistent with the fact that some earlier modeling studies needed an arbitrary increase in BC vehicular emissions to match observations. The fact that their comparisons with both BC air concentrations and deposited mass are consistent is particularly interesting, because it provides closure to the air pollution process being modeled.

In addition, this work confirms the importance of treating urban air quality at various scales in a joint consistent manner. Previous work by the authors and others had demonstrated this point for reactive gaseous pollutants. It is demonstrated here for chemically inert particulate matter.

Therefore, I recommend publication of this work in GMD with minor modifications to address the following comments. Also, the text needs some editing by the authors to correct some English mistakes and improve the style overall.

Specific comments

The conclusion that non-exhaust BC emissions are currently underestimated in European emission inventories (according to the results presented here) has important policy implications. It means that replacing internal-combustion engines by electric vehicles will not eliminate BC air pollution, but only reduce it by about half (in terms of vehicular emissions). This point should be highlighted, both in the conclusion and in the abstract. Clearly, additional work (mostly experimental) should be conducted to confirm this result, but the work presented here makes a very strong case for revising current BC vehicular emission inventories and taking non-exhaust emissions explicitly into account in air quality simulations. Some recommendations regarding additional studies to refine those results could be provided in the conclusion.

The introduction is rather long (about one quarter of the entire text) and the authors should consider breaking this current introductory section into two sections: (1) a general introduction and (2) a section that provides an overview of the current state of the science for non-exhaust vehicular emissions (since it corresponds to the main topic of this work).

In the first paragraph of the introduction, it is stated that BC is mainly emitted by traffic. It should be pointed out that during winter, wood burning (for residential heating) is also an important source of BC in some areas (including the Paris region).

Among the potential health effects of BC, should its carcinogenic effects be mentioned (e.g., Lequy et al., Environ. Health Perspectives, 129, March 2021)?

The sentence on the environmental impacts of BC needs to be rewritten, because visibility reduction is due to the radiative effects of BC. I suggest: "... environmental impacts due to its radiative properties (light absorption), which lead to visibility reduction (references) and global warming (references).

p. 2, line 49: delete "in models" (the under-estimation affects the emission inventory in general, which may be used for modeling or other tasks, such as reporting).

p. 3, line 82 and others: the unit mg.vkm-1 is used throughout (milligrams per vehiclekilometer travelled); however, later in the text (e.g., p. 9, line 255) the unit veh.h-1 (vehicles per hour) is used for traffic flow. Unit notations should be consistent for "vehicles". I suggest using mg.(veh-km)-1 for the former.

On p. 4, lines 107-109, it is mentioned that the road-wear emissions calculated by Thouron et al. (2018) were greater than those of the EMEP guidelines. Can the authors specify whether this is due to the algorithm used or the road characteristics (or a combination of both)?

p. 4, line 105: I think that "soil" would only apply to a dirt road. I suggest "road characteristics" instead (also p. 13, line 360).

p. 5, line 133: "worst quality codes"; I am not sure what the authors mean.

In Sections 2 and 3, it is mentioned that particulate BC is treated using a sectional size distribution with 6 size sections, which is useful to correctly simulate dry deposition. Since atmospheric chemistry is not treated, I assume that aerosol dynamics (condensation/evaporation, nucleation, and coagulation) is also not treated. This seems appropriate as it would have little effect on the BC size distribution (in particular, considering the uncertainties associated with its initial size distribution). The authors should mention this point. Also, the initial size distribution of emitted particulate BC should be specified, either with a reference or by providing the size distribution.

p. 10, line 280: it is not clear why $Q_{emis, exh}$ represents both exhaust and non-exhaust emissions. Please clarify.

p. 10, line 280: it is not clear in $Q_{dep, v}$ what the second subscript, v, represents. Please clarify.

In Section 3, p. 12, Equation 10: please specify the units of EF.

In Section 4.1, p. 13, line 343: "to estimate and control"; please specify what "control" refers to.

At the end of Section 4.1, the authors mention that their correction factor for exhaust BC emissions used in simulation 6 is based on traffic flow characteristics and BC/PM2.5 uncertainties. Could this be explained in greater detail?

In Section 4.2, the authors provide two different sets of criteria for model performance evaluation. The stricter set includes 6 performance metrics (5 were initially defined by Hanna et al., Atmos. Environ., 38, 4675-4687, 2004; the normalized absolute difference was added later), whereas the less strict set (suitable for urban areas according to Hanna and Chang, 2012) includes only 4. It is not clear whether the authors consider that the stricter criteria defined for MG and VG should also apply to the less strict set or whether those metrics must simply be dropped from the less strict set; this point should be clarified. It may be a moot point since, in any case, simulations 1 through 3 fail the less strict criteria (except NMSE) and simulations 4 through 6 meet all the stricter criteria.

In Section 4.3, p. 17, lines 425-426: I would not say that the observations of Amato et al. are "quite similar" because the result obtained in this work is slightly above the upper bound of the range given by Amato et al.

In Section 4.3, the authors refer to PM10 as fine particles. This definition of PM10 is unfortunately commonly used in France by some agencies, although it is incorrect. Fine particles correspond to PM2.5 (give or take 1 ïA■m) and PM10, therefore, includes fine particles and a fraction of coarse particles.

In Section 5, p. 19, line 477: "lower concentrations"? Should it be "higher concentrations" if there is double counting?

As mentioned above, the authors should mention the policy implications of their results concerning BC emissions from vehicles in the conclusion. Also, some suggestions for additional experimental studies to confirm the results of their work would be appropriate.

---

## Author Comment (AC1) · 27 Aug 2021

article [utf8]inputenc [a4paper, total=6in, 8in]geometry xurl comment
**Black carbon modelling in urban areas: investigating the influence of resuspension and non-exhaust emissions in streets using the Street-in-Grid (SinG) model :**
**Answers to referees' comments**

August 27, 2021

**1   Executive editor - Astrid Kerkweg**

Dear authors,

in my role as Executive editor of GMD, I would like to bring to your attention our Editorial version 1.2: https://www.geosci-model-dev.net/12/2215/2019/ This highlights some requirements of papers published in GMD, which is also available on the GMD website in the 'Manuscript Types' section: http://www.geoscientific-model-development.

net/submission/manuscript_types.html

In particular, please note that for your paper, the following requirement has not been met in the Discussions paper:

- "The main paper must give the model name and version number (or other unique identifier) in the title."

Please add a version number for SinG in the title upon your revised submission to GMD.

Yours,

Astrid Kerkweg

*Author response:* The SinG version is now included in the title: *"Black carbon modelling in urban areas: investigating the influence of resuspension and non-exhaust emissions in streets using the Street-in-Grid model for inert particles (SinG-inert)".*

---

## Author Comment (AC2) · 27 Aug 2021

article [utf8]inputenc [a4paper, total=6in, 8in]geometry xurl comment

Black carbon modelling in urban areas: investigating the influence of resuspension and non-exhaust emissions in streets using the Street-in-Grid (SinG) model :
**Answers to referees' comments**

August 27, 2021

**1  Anonymous Referee #1**

**1.1  General comments**

The underestimation of BC concentration in air quality models has been a comment issue and this work provides some perspectives that would be important to solve this problem. Those founding are important and could be of the interest of readers and scientific communities in this field. However, the languages and overall structure of the

manuscript should be improved before publications. My comments are as follows:

**1.2   Major comments**

1.  The introduction is too long and requires serious streamlining.  I would suggest the author summarize those modeling literature and emission numbers using one or two tables.  And only discuss the technical difference between those works.  Also, to make the overall structure more clear, the authors could consider rearranging the literature part based on the following order: observations, modeling methods, model performance, gaps need to be addressed in current models.

*Author response:* Yes, we agree that the introduction was too long.  Following your comment and also the second referee suggestion, we added a new section (*Uncertainties and variability of emissions factors observed in non-exhaust emissions*).  This section is included after the introduction, and include the discussions about the variability observed on non-exhaust emissions in the literature.

2. The description of different simulations conducted in this study is confusing.  After careful and difficult reading, I guess there are two different sets of simulations.  One set is 6 simulations with Polairs3D-SinG-MUNICH model, another one is 1 simulation with Polair3D-MUNICH model. If that is the case, the author should make it more clear in Section 2.  Also, the author should improve the simulation scenario description by adding key massages in Table 1, such as the model/source used for wear emission calculation.

*Author response:* Different modifications were made to improve clarity, following the different comments of the reviewer. It is now specified in the Section 2 (*Model description*) that MUNICH can be used as a stand-alone model.  This enables a better link between the Model description and the Section 5 *Influence of the two-way coupling*

*on BC concentrations in streets* (as also asked in the minor comment number 17). It is also highlighted that all the simulations presented in Section 4.1 *The simulations* were performed using SinG (minor comment number 13), and we added in Table 1 the reference corresponding to each test (minor comment number 15).

3. The author should explain why the simulation results present in Section 5 are not evaluate with observations. Without the model performance compared, it is impossible to evaluate the necessity or advantage to adopt the two-way coupling method compares to the one-way coupling method.

*Author response:* As explained in the reply to the minor comment 21, both SinG and MUNICH used as a stand-alone model (MUNICH-only) result in very similar concentrations at Boulevard Alsace Lorraine, and they both respect the most strict performance criteria. The comparisons between BC concentrations observed at Boulevard Alsace Lorraine and those calculated with SinG and MUNICH is now included. The influence of the two-way coupling depends on the street characteristics, it is more important in other streets than Boulevard Alsace Lorraine. However, there was no measurement in those streets.

Because the uncertainties of non-exhaust emissions are still high, it is difficult to evaluate with the measurements which coupling approach has better performance. As mentioned in the Conclusion (lines 500-502), *"Non-exhaust emissions still present very high uncertainties, with a large spectrum of emission factors and experimental limitations to separate each non-exhaust emission source. More studies are needed to better characterize these emissions, their size distributions and chemical compositions".*

4. The author takes the entire section 2 to describe a new "deposition-resuspension" framework but concludes that the whole "deposition-resuspension" process is insignificant in improving BC underestimation (line 405-410). Instead, the selection of emission factor is the key to solve the problem. This made the work of the new "deposition-resuspension" framework meanless. The author could add one more simulation with the setting of simulation 4 while turn off the new "mass-conserved deposition-resuspension" framework, to properly evaluate the potential of this framework. If not, I would suggest the author move entire section 2 to the appendix and only briefly introduce the new framework in the "Simulation setup" section.

*Author response:* Yes, the particle resuspension calculated is this study does not strongly affect the black carbon concentration in the streets, whereas tyre-wear emissions do. This is a strong conclusion, even though it is negative (resuspension is insignificant for BC). Therefore, it seems important to accurately describe the approach used to calculate resuspension to reinforce the conclusions. Also, this approach to calculate particle resuspension is innovative and realistic, in the sense that it calculates particle resuspension as a function of the available mass on the street surface.

The simulation 5 in this study uses the setting of simulation 4, turning off the new "mass-conserved deposition-resuspension" framework. To improve clarity, the sentences

*"In order to evaluate the impact of particle resuspension on BC concentrations in streets, simulation 5 is similar to simulation 4, but it does not take into account particle deposition. In other words, simulation 5 simulates a maximal resuspension rate, equals to the deposition rate."*

are replaced by

*"In order to evaluate the impact of particle resuspension on BC concentrations in streets, simulation 5 uses the same setting of simulation 4, but it does not take into account particle deposition, and then, particle resuspension is not computed. This simulation physically represents the concentrations obtained with a maximal BC resuspension, which is equal to the BC deposition.".*

5. In general, the languages used in this manuscript could be improved. Try to avoid long sentences with multiple subordinate clauses that really made the text difficult to follow.

***Author response:*** The language was improved, and long sentences were avoided in all the modifications indicated here to answer the comments. In addition to these modifications, some sentences were also reformulated:

[revised manuscript text omitted]

1.3   Minor comments

1. Line 7-9, the statement after "i.e." is confusing, please rephrase. Do you mean the BC from one street can be transported to another street at regional scales?

***Author response:*** We mean that the high concentrations of BC in a street can influence the urban background concentration above that street.

To improve clarity, the sentence

"*In terms of modelling, the street models do not always consider the two-way interactions between the local and regional scales, i.e. the influence of the high BC concentrations observed in streets on the urban background concentrations, which can enhance the BC concentrations in streets.*"

is replaced by

"*In terms of modelling, the street models do not always consider the two-way inter-*

*actions between the local and regional scales. With the two-way modeling approach, a street with high BC concentrations may influence urban background concentrations above that street, which can finally enhance the BC concentrations in the same street.*".

2. Line 22-23: Please rephrase. For example, Here comparisons are performed using ... factors from literature, and we found ?? are improved?

*Author response:* The sentences

"*Here, emission factors of tyre, brake and road wear are calculated based on the literature, and a sensitivity analysis of these emission factors on BC concentrations in streets is performed. The model to measurement comparison shows that tyre-emission factors usually used in Europe are probably under-estimated, and tyre-emission factors coherent with some studies of the literature and the comparison performed here are proposed.*"

are replaced by

"*Here, a sensitivity analysis of BC concentrations is performed by comparing simulations with different emission factors of tyre, brake and road wear. The different emission factors considered here are estimated based on the literature. We found a satisfying model-to-measurement comparison using high tyre-wear emission factors, which may indicate that tyre-emission factors usually used in Europe are probably under-estimated.* " .

3. Line 58: Can you provide the LDV and HDV definition (citation?), why the MDV (mid-duty vehicles) is not considered in this study?

*Author response:* In the available traffic flow observations performed during the Trafipollu campaign, and used in this study, the vehicles are classified as HDV and

LDV. The same classification is used in other European studies, as in the NORTRIP model, defining non-exhaust emission factors only for HDV and LDV vehicle categories. HDV and LDV are classified according to the European Emission guidelines (EMEP), with light commercial vehicles lighter than 3.5 tons, and heavy-duty vehicles, heavier than 3.5 tons. Differently, the US vehicle classification also contains the MDV vehicle class, with vehicles heavier than 4.25 tons and lighter than 5 tons.

To improve clarity, the sentence

*"In Europe, exhaust emission factors are nowadays determined according to the vehicle technology and fuel, providing realistic emission factors for divers vehicle fleet, as detailed in the EMEP guidelines (Ntziachristos and Samaras, 2018)."*

is replaced by:

*"In Europe, exhaust emission factors are nowadays determined according to the vehicle technology and fuel, providing realistic emission factors for divers vehicle fleet. The details about vehicle categories, technologies and fuel are detailed in the EMEP guidelines (Ntziachristos and Samaras, 2018)."*

4. Line 75-80: Feels like most of the non-exhaust important studies are $PM_{10}$ focused. Can you discuss the size distribution difference between exhaust and non-exhaust emissions?

***Author response:*** Yes, most of the non-exhaust important studies are $PM_{10}$ focused. While exhaust emissions present tiny particles, lower than 1 $\mu$m ($PM_1$), non-exhaust emissions present, in general, coarser particles, mostly higher than 2.5 $\mu$m ($PM_{2.5}$).

The sentence

*"They estimated non-exhaust emissions to be approximately of the same magnitude as exhaust emissions."*

is replaced by:

*"They estimated that the mass of coarse particles ($PM_{10-2.5}$) from non-exhaust emissions is approximately the same as the mass of fine particles ($PM_{2.5}$) from exhaust emissions. Note that important differences are observed in their size distribution. Exhaust emissions are composed of tiny particles, with diameters lower than 1 $\mu m$ (Ntziachristos and Samaras, 2018). Particles from non-exhaust emissions are coarser. According to the European Emission guidelines (Ntziachristos and Samaras, 2016), 60% of particles emitted from tyre wear are $PM_{10}$, 42% are $PM_{2.5}$, and only 6% are $PM_1$. Particles from brake-wear emissions have lower diameters: 98% are $PM_{10}$, 39% are $PM_{2.5}$, and 10% are $PM_1$. The coarsest particles are those from road wear: 50% are $PM_{10}$, 27% are $PM_{2.5}$, and there is no $PM_1$."*

5. Line 84-85: Is it the same ratio for both $PM_{2.5}$ and $PM_{10}$?

***Author response:*** No, the authors consider that exhaust emission correspond to $PM_{2.5}$, while non-exhaust emissions correspond to coarse particles, between $PM_{2.5}$ and $PM_{10}$.

To improve clarity, the sentence

*"They estimated non-exhaust emissions to be approximately of the same magnitude as exhaust emissions."*

is replaced by

*"They estimated that the mass of coarse particles ($PM_{10-2.5}$) from non-exhaust emissions is approximately the same as the mass of fine particles ($PM_{2.5}$) from exhaust emissions."*

6. Line 87: How is this related to non-exhaust emission? What is the source of this

species?

*Author response:* According to Khare et al. (2020), the asphalt present in the road pavement is an important source of organic compounds. In determined temperatures and solar radiation intensities, the asphalt evaporates organic compounds during its' different life cycle stages (such as storage, paving, and use).

To improve clarity, the sentence

*"Khare et al. (2020) highlighted the importance of asphalt-related emissions in SOA formation, still absent from emission inventories."*

is replaced by

*"Khare et al. (2020) highlighted the importance of asphalt-related emissions in SOA formation, still absent from emission inventories. These emissions are dependent of the solar radiation and temperature over the asphalt surface, and are variable according to the asphalt life cycle stages (such as storage, paving, and use).".*

7. Line 91: "Some models" Please provide citations. And also, I don't think even if is appropriate here, most would be a better wording.

*Author response:* To use a better wording and add the citations, the sentence

*"Some models try to integrate vehicle operational conditions to estimate non-exhaust emission factors, even if based on empirical factors and simplified parametrizations"*

is replaced to

*"A few emission models try to integrate vehicle operational conditions to estimate non-exhaust emission factors, based on empirical factors and simplified parameterizations (Guevara et al., 2020, Denby et al., 2013a)".*

8. Line 92: "HERMES" Please provide a citation of this model here.

*Author response:* The sentence

*"The HERMES model employs the non-exhaust emission factors proposed in the EMEP guidelines (Ntziachristos and Boulter, 2016), which provide $PM_{10}$ wear emission rates for passenger cars, motorcycles, LDV and HDV."*

is replaced by

*"The High-Elective Resolution Modelling Emission System (HERMES) model (Guevara et al, 2020) employs the non-exhaust emission factors proposed in the EMEP guidelines (Ntziachristos and Boulter, 2016), which provide $PM_{10}$ wear emission rates for passenger cars, motorcycles, LDV and HDV.".*

Please, note that HERMES model reference at line 356 was updated. The article, previously in revision, is now published.

9. Line 103: Why is it supposed to be independent? Are describing the NORTRIP assumption or making general claims? Please clear, if it is the latter, you would need a citation to justify your claims.

*Author response:* We describe the NORTRIP model assumption. Equation 10 in Denby et al. (2013a) describes their parameterization to calculate brake wear emissions, and it is independent of the vehicle speed.

To improve clarity, the sentence

*"Differently, the NORTRIP model takes into account the vehicle speed to determine $PM_{10}$ tyre and road wear emission rates, but brake-wear emissions are supposed to be independent of vehicle speed."*

is replaced by

*"Differently, the NORTRIP model takes into account the vehicle speed to determine $PM_{10}$ tyre and road wear emission rates, but not for brake-wear emissions, which are assumed to be independent of vehicle speed."*

10. Line 317: Why selecting those two days? Are they more representative of the general average?

***Author response:*** Exhaust emissions were calculated in the Trafipollu project (https://anr.fr/Projet-ANR-12-VBDU-0002), and then used by the authors. These two days were previously defined by the participants of Trafipollu project. To validate this approach, the participants of Trafipollu project analyzed the variability of traffic and daily hourly profiles at Boulevard Alsace Lorraine on weekdays and weekdays. Observations regarding workdays were performed from 31 March 2014 to 04 April 2014, and those regarding weekends were performed at 29 and 30 March 2014, and 5, 6, 12 and 13 April 2014. Low variations of total traffic was observed during workdays (around 10%). Slightly higher variations are observed on weekends (around 20%), but the total traffic in these days are considerably lower than in workdays (approximately the half of total traffic in workdays). These low variations in total traffic validate the approach of using two representative days to simulate pollutant emissions. Other studies also mention the low variations in traffic emissions during the year, as the Source apportionment of airbone particles in the Île-de-France region, performed by the Air-quality agency of Paris (AIRPARIF, 2012).

Note that the Trafipollu web page is updated to https://anr.fr/Projet-ANR-12-VBDU-0002.

11. Line 324: "$PM_{10}$ fraction" fraction of what? $PM_{10}$ fraction of TSP?

***Author response:*** Yes. To improve clarity, the sentence

*"$f_s^{pm10}$ the PM$_{10}$ fraction in each wear source $s$"*

is replaced by

*"$f_s^{pm10}$ the PM$_{10}$ fraction of TSP emitted by each wear source $s$".*

12. Line 330: What is LCFs stand for by each letter?

***Author response:*** It means load correction factor.

To improve clarity, the sentence

*"They take into account the vehicle characteristics as HDV weight ($LCF_s$) and the number of axles ($N_{axle}$), as indicated in Equations (14) and (15)."*

is replaced by

*"They take into account the vehicle characteristics as HDV weight (represented by a load correction factor $LCF_s$) and the number of axles ($N_{axle}$), as indicated in Equations (14) and (15)."*

13. Section 4.1: Not clear if those 6 simulations are conducted based on Polairs3D-MUNICH-SinG model combination or on a simple SinG local/box model.

***Author response:*** All simulations in Section 4.1 were made using the SinG model. This model performs a two-way dynamic coupling between the regional-scale model Polair3D, and the local-scale model MUNICH, and can be seen as an interface between these models. With the two-way dynamic approach, the concentrations calculated in streets affect the concentrations calculated in the urban backgroud, and vice-versa.

Note that MUNICH can be also used as a stand-alone model, with a one-way coupling approach between regional and local scales. With this one-way coupling approach,

the concentrations calculated in the streets do not affect the concentrations calculated in the urban background. The comparisons between the two-way (SinG) and one-way (MUNICH as a stand-alone model) coupling approaches is performed in Section 5.

To improve clarity, the sentence

*"Different simulations were performed, with large variations in tyre-wear emissions properties."*

is replaced by

*"Different simulations were performed using SinG, with large variations in tyre-wear emissions properties."*.

14. Line 350: How dose those fractions applied for different size bins?

***Author response:*** These fractions are used as constants, as indicated in the European Emission guidelines (Ntziachristos and Boulter, 2016).

To improve clarity, the sentence

*"The BC fraction adopted in brake and road wear are constant in all simulations, and follow the EMEP guidelines (Ntziachristos and Boulter, 2016) with $f_{bk}^{bc}$ = 0.026 and $f_{rd}^{bc}$ = 0.0106."*

is replaced by

*"The BC fractions adopted in brake and road wear are constant in all simulations and they are the same in all emitted size sections. They follow the EMEP guidelines (Ntziachristos and Boulter, 2016) with $f_{bk}^{bc}$ = 0.026 and $f_{rd}^{bc}$ = 0.0106."*

15. Table 1: Can you also add the model SOURCE in the table? Like EMEP, HERMES,

and NORTRIP?

***Author response:*** Yes, it was added as footnote.

16: Line 395-398: I don't understand here. If the non-exhaust wear emissions are very low, there should be a very small amount of mass available for resuspension. So, no matter how high the resuspension rate is, there should have no mass supply for resuspension, as demonstrated by simulation 5. So, how could a high resuspension rate along explained the much higher BC concentration in previous simulations? Isn't it just contradict your simulation 5?

***Author response:*** The previous simulations mentioned in line 395 refer to Denby et al. (2013a) and Pay et al. (2011), not to simulations 2 and 3 of this study. Even if simulations 2 and 3 of this study use the same tyre, brake and road wear emission factors than those used by Pay et al. (2011) and Denby et al. (2013a), respectively, the parameterization employed to calculate particle resuspension are different. Pay et al. (2011) use constant resuspension emission factors, wich may not respect the mass balance over the street surface. Denby et al. (2013a) calculates the mass balance over the street surface, but artificially assuming that tyre, brake and road-wear emissions are instantly deposited over the street surface, and only these sources are employed to calculate the particle mass on the street. In both cases the authors found a relevant contribution of particle resuspension, and employed relatively low values of tyre-wear emission factors. Differently, in all simulations performed in this study, particle resuspension is computed by strictly respecting the mass balance over the street surface. Here the available mass at the street surface is computed based on particle deposition in the atmosphere, including all different sources, and particle wash-off, in rain episodes. With this approach, the particle resuspension calculated for black carbon does not strongly affect the concentrations in the street. These different approaches to calculate particle resuspension can explain the differences between the conclusions
obtained here and those obtained by Pay et al. (2011) and Denby et al. (2013a), as discussed at lines 398-403.

To improve clarity, the sentence

*"Previous simulations with the NORTRIP and HERMES models (Denby et al., 2013a; Pay et al., 2011) achieved good correlations between simulated and measured particle concentrations, because they assume that resuspension is the main non-exhaust emission process and use high resuspension rates."*

is replaced by

*"The simulations performed by Pay et al. (2011) and Denby et al. (2013a), using HERMES and NORTRIP models respectively, achieved good correlations between simulated and measured particle concentrations. They assume that resuspension is the main non-exhaust emission process, using high resuspension rates and relatively low tyre-wear emission factors.".*

17. Line 442: Very confusing statement here, do you mean SinG and MUNICH are two different systems that can replace each other? But based on section 2, MUNICH is part of SinG. Or, maybe you want to say, the comparison is between Polair3D-SinG-MUNICH and Polair3D-MUNICH (without SinG)? Same question for line 465.

***Author response:*** The local-scale model MUNICH is part of SinG, but it can also be used as a stand-alone model. The different between these two approaches is the interactions between local (streets) and regional (urban background) scales.

As explained in lines 442-452: *"the coupling between the local and regional scales is two way in SinG, which couple the street model MUNICH to the chemistry transport model Polair3D. However, MUNICH may be used as a standalone model, simulating the street concentrations with a one-way coupling to the regional-scale background concentrations. In that case, the regional-scale (background) concentrations influence*

*the street concentrations, but the street concentrations do not influence the background concentrations, and the vertical mass transfer between local and regional scales only influences concentrations in the streets. Note that in the one-way coupling, traffic emissions are used both in the regional-scale model Polair3D and in the street model MUNICH. Because SinG employs a two-way coupling, at each time step the vertical mass transfer between local and regional scales enables to calculate concentrations in the background and in streets, providing a direct interaction between concentrations in the street network and those in the urban background. Therefore, traffic emissions are considered only at the local scale, and there is no double counting of traffic emissions in SinG."*

To improve clarity, these information are added and now briefly mentioned in Section 2. At line 204, the sentences

*"As detailed in Lugon et al. (2020), this dynamic coupling between local and regional scales allows a direct interaction between concentrations in the street network and those in the urban background: the mass transfer between the street and the background concentrations influence both the street and the background concentrations. Furthermore, SinG uses consistent chemical and physical parameterizations, such as the same chemical module and meteorological data, at both local and regional scales."*

are followed by

*"Note that the street-network model MUNICH can also be used as a stand-alone model, with a one-way coupling approach. In this case the background concentrations influence the concentrations in the streets, but the mass transfer between streets and the background do not influence background concentrations.".*

18. Line 444-448: What is the point to mention the one-way feedback here? Is it used in this study? Do you mean the Polair3D-SinG-MUNICH is using two-way feedback,

and Polair3D-MUNICH is using one-way feedback? And you are comparing between those two methods?

*Author response:* Yes, exactly. This section compares the two approaches to calculate black carbon concentrations in the streets. Two simulations are performed using the same inlet data: $(i)$ one simulation uses SinG, with the two-way coupling approach, and $(ii)$ the other simulation using MUNICH as a stand-alone model, with the one-way coupling approach.

To increase clarity, in addition to the notion of MUNICH used as a stand-alone model in Section 2 (please, see comment 17), the sentence

*"This section investigates the influence of the two-way coupling between the regional and local scales on BC concentrations in the street network by comparing the concentrations simulated by SinG and MUNICH."*

is replaced by

*"This section investigates the influence of the two-way coupling between the regional and local scales on BC concentrations in the street network. For this, the concentrations calculated by SinG, with the two-way approach, are compared with those calculated by MUNICH as a stand-alone model, with the one-way approach."*

19. Line 469-470: Can you explain why this SinG results to higher BC concentrations compares to MUNICH for those streets?

*Author response:* As mentioned in line 470, these streets present high traffic emissions. According to the explanations of lines 456-462, *"The vertical mass flux between the local and regional scales is proportional to the concentration gradient between the street and the urban background, as shown in Equation (8) of Lugon et al. (2020). Streets with high traffic emissions tend to favor the vertical mass transfer from the local to the regional scales, as they tend to present a high gradient between the street*

*and the urban background concentrations. This vertical mass flux is also dependent of the street geometry, represented by the aspect ratio $\alpha r$ (see Equation 9 in Lugon et al. (2020)). Streets with low aspect ratio (large streets) tend to favor the vertical transfer between local and regional scales, and in streets with high aspect ratios (narrow streets) the vertical mass transfer between scales tends to be lower.*

20. Line 477-478: How do this double-counting works? Please provide more detailed discussion here. Why does this not seeing on high emission street?

*Author response:* The double-counting concept is mentioned in lines 447-452, as follows:

*"Note that in the one-way coupling, traffic emissions are used both in the regional-scale model Polair3D and in the street model MUNICH. Because SinG employs a two-way coupling, at each time step the vertical mass transfer between local and regional scales enables to calculate concentrations in the background and in streets, providing a direct interaction between concentrations in the street network and those in the urban background. Therefore, traffic emissions are considered only at the local scale, and there is no double counting of traffic emissions in SinG."*

This approach is a simplified technique to represent the influence of the street network on background concentrations in the one-way coupling approach. Differently, with the two-way coupling approach, the mass transfer between streets and urban background is considered at both scales, and then there is no need to consider the traffic emissions at the regional scale. In streets with high traffic emissions, the concentration gradient favors the mass transfer between local and regional scales, and this mass flux is higher than the traffic emissions considered in the one-way coupling approach. In streets with low traffic emissions and high aspect ratios, it is the opposite: the vertical mass transfer between streets and urban background is low. The traffic emissions considered at the regional scale using the one-way coupling approach is then higher than the mass

transfer between streets and urban background considered in the two-way coupling approach. In this case, this results in higher background concentrations above the streets using the one-way coupling approach, leading then to higher street concentrations.

To improve clarity, the sentence

*"These large differences and the lower concentrations simulated in MUNICH compared to SinG may be explained by the double counting of traffic emissions performed by MUNICH with the one-way coupling technique."*

is replaced by

*"These large differences and the lower concentrations simulated in SinG compared to MUNICH may be explained by the double counting of traffic emissions performed by MUNICH with the one-way coupling technique. Using the two-way coupling approach in SinG, the streets with low traffic emissions and high aspect ratios present low vertical mass transfer between streets and urban background. This mass transfer is lower than the traffic emissions considered at the regional scale in the one-way coupling approach (MUNICH). This results in lower background concentrations above the streets using the two-way coupling approach (SinG), leading then to lower street concentrations. "*

21. Section 5: Is the SinG simulation in Section 5 the same as simulation 4 in section 4? Is the MUNICH simulation in section 5 have the same model setup for all other SinG simulations? Why simulation result from the MUNICH simulation not compared with street-level observation (Boulevard Alsace Lorraine) as in Section 4? There is no way to judge which method has better model performance MUNICH or SinG.

***Author response:*** Yes, the simulation using MUNICH as a stand-alone model (MUNICH-only) used the same input data as the simulation 4 presented in Section 4. Both MUNICH-only and SinG result very similar concentrations at Boulevard Alsace Lorraine, and they both respect the most strict performance criteria. The comparisons

between BC concentrations observed at Boulevard Alsace Lorraine and those calculated with SinG and MUNICH are now included. The statistical indicators comparing SinG and MUNICH results are presented in the Appendix. For this, at line 453, the sentence

*"Simulations of BC concentrations using Polair3D, MUNICH and SinG are performed using the non-exhaust emission factors of simulation 4 (see Table 1)."*

is followed by

*"Regarding specifically the BC concentrations at the Boulevard Alsace Lorraine, both SinG and MUNICH as a stand-alone model (MUNICH-only) resulted in similar concentrations, respecting the most strict performance criteria (see Table A1 in Appendix A2). However, the differences between MUNICH and SinG depend on the street characteristics, as detailed in Lugon et al (2020). To analyze streets with different characteristics, this section focus on the comparison between the one-way and two-way coupling approaches over the whole street network.".*

And the following Table is included in the Appendix section:

**Table 1.** Comparisons to BC measurements at "Boulevard Alsace-Lorraine": statistical indicators obtained with SinG and MUNICH-only simulations.

|  | o [$\mu$g.m$^{-3}$] | s [$\mu$g.m$^{-3}$] | FB | MG | NMSE | VG | FAC2 | NAD |
|---|---|---|---|---|---|---|---|---|
| SinG | 6.07 | 4.91 | -0.21 | 0.82 | 0.29 | 1.27 | 0.77 | 0.19 |
| MUNICH-only | 6.07 | 4.97 | -0.19 | 0.82 | 0.33 | 1.32 | 0.74 | 0.20 |

---

## Author Comment (AC3) · 27 Aug 2021

article [utf8]inputenc [a4paper, total=6in, 8in]geometry xurl comment
**Black carbon modelling in urban areas: investigating the influence of resuspension and non-exhaust emissions in streets using the Street-in-Grid (SinG) model :**
**Answers to referees' comments**

August 27, 2021

**1  Anonymous Referee #2**

**1.1  General comments**

This research article provides new and very useful information regarding the contribution of on-road traffic to urban air pollution. The focus is on particulate black carbon (BC) and its vehicular emission sources. The authors use a novel advanced modeling technique, the SinG model, which simulates urban air pollution at various spatial scales

(urban background and street-level pollution) in an internally consistent manner. Furthermore, they improved this model by implementing a more detailed dry deposition model and conducting a dynamic simulation of the emission, deposition, and resuspension of particulate matter.

In this work, they apply this model to investigate the vehicular emissions of BC. In particular, they evaluate the relative contributions of exhaust and non-exhaust (tire, brake, and road wear) emissions of BC to air pollution in streets. The non-exhaust emission source is particularly uncertain and the authors do an excellent job at reviewing the literature, identifying the major uncertainty sources, and systematically investigating their impact on simulated BC concentrations. They conduct a thorough evaluation of the modeling results using experimental results of a field program that provides both ambient air BC concentrations in a Paris suburban street and deposited PM mass in that same street. This evaluation allows the authors to propose that the greater estimates of non-exhaust emissions are more realistic and that current non-exhaust BC emission inventories may significantly underestimate actual BC vehicular emissions.

This conclusion is consistent with the fact that some earlier modeling studies needed an arbitrary increase in BC vehicular emissions to match observations. The fact that their comparisons with both BC air concentrations and deposited mass are consistent is particularly interesting, because it provides closure to the air pollution process being modeled. In addition, this work confirms the importance of treating urban air quality at various scales in a joint consistent manner. Previous work by the authors and others had demonstrated this point for reactive gaseous pollutants. It is demonstrated here for chemically inert particulate matter.

Therefore, I recommend publication of this work in GMD with minor modifications to address the following comments. Also, the text needs some editing by the authors to correct some English mistakes and improve the style overall.

**1.2 Specific comments**

1. The conclusion that non-exhaust BC emissions are currently underestimated in European emission inventories (according to the results presented here) has important policy implications. It means that replacing internal-combustion engines by electric vehicles will not eliminate BC air pollution, but only reduce it by about half (in terms of vehicular emissions). This point should be highlighted, both in the conclusion and in the abstract. Clearly, additional work (mostly experimental) should be conducted to confirm this result, but the work presented here makes a very strong case for revising current BC vehicular emission inventories and taking non-exhaust emissions explicitly into account in air quality simulations. Some recommendations regarding additional studies to refine those results could be provided in the conclusion.

***Author response:*** Yes, thank you. These points are now highlighted. At lines 19-23, the sentences

*"Non-exhaust emission, such as brake and tyre and road wear, largely contribute to BC emissions, with a contribution equivalent to exhaust emissions. Here, emission factors of tyre, brake and road wear are calculated based on the literature, and a sensitivity analysis of these emission factors on BC concentrations in streets is performed. The model to measurement comparison shows that tyre-emission factors usually used in Europe are probably under-estimated, and tyre-emission factors coherent with some studies of the literature and the comparison performed here are proposed."*

are replaced by

*"Non-exhaust emissions, such as brake and tyre and road wear, may largely contribute to BC emissions, with a contribution equivalent to exhaust emissions. Here, a sensitivity analysis of BC concentrations is performed by comparing simulations with different emission factors of tyre, brake and road wear. The different emission factors considered are estimated based on the literature. We found a satisfying model-to-*

measurement comparison using high tyre-wear emission factors, which may indicate
that tyre-emission factors usually used in Europe are probably under-estimated. These
results have important policy implications: public policies replacing internal-combustion
engines by electric vehicles may not eliminate BC air pollution, but only reduce it by
half."

Also, at line 409, the sentence

"They can be as relevant as exhaust emissions, and their underestimation may justify the virtual increase of BC emissions often employed in street-network modelling
studies."

is followed by

"This can have direct consequences on public policies aiming at reducing BC concentrations in urban areas: non-exhaust emissions may contribute to as much as half of
the BC concentrations in streets.".

Finally, at line 501, the sentence

"Following the literature, increasing the BC passenger cars tyre-wear emission factors
of the EMEP guidelines from 1.36 mg.vkm$^{-1}$ to 20.8 mg.vkm$^{-1}$ lead to good comparisons of the simulated BC concentrations to the measured ones."

is followed by

"The authors highlight that additional work, mostly experimental, should be conducted
to confirm this result. However, the work presented here makes a very strong case for
revising current BC vehicular emission inventories and taking non-exhaust emissions
explicitly into account in air quality simulations."

2. The introduction is rather long (about one quarter of the entire text) and the authors
should consider breaking this current introductory section into two sections: (1) a general introduction and (2) a section that provides an overview of the current state of the science for non-exhaust vehicular emissions (since it corresponds to the main topic of this work).

*Author response:* As suggested, the Introduction was divided in two sections: Introduction and Uncertainties and variability of emissions factors observed in non-exhaust emissions. The Introduction starts at line 25, until line 64. At line 64, the sentence

[revised manuscript text omitted]

7. p. 3, line 82 and others: the unit mg.vkm$-1$ is used throughout (milligrams per vehicle-kilometer travelled); however, later in the text (e.g., p. 9, line 255) the unit veh.h$-1$ (vehicles per hour) is used for traffic flow. Unit notations should be consistent for "vehicles". I suggest using mg.(veh-km)$-1$ for the former.

***Author response:*** It was corrected.

8. On p. 4, lines 107-109, it is mentioned that the road-wear emissions calculated by Thouron et al. (2018) were greater than those of the EMEP guidelines. Can the authors specify whether this is due to the algorithm used or the road characteristics (or a combination of both)?

***Author response:*** It is due to the combination of both, algorithm and road characteristics (represented by the $h_{pavement}$).

At lines 107-109, the sentence

*"However, the road-wear emissions calculated using the NORTRIP model with the soil characteristics employed by Thouron et al. (2018) in "Boulevard Alsace Lorraine" (a street East of Paris) are higher than those proposed in the EMEP guidelines, by a ratio 6.0."*

is replaced by

*"The road-wear emissions calculated using the NORTRIP model with the road characteristics employed by Thouron et al. (2018) in "Boulevard Alsace Lorraine" (a street East of Paris). They are higher than those proposed in the EMEP guidelines, by a ratio 6.0, because of their different algorithm and road characteristics.".*

9. p. 4, line 105: I think that "soil" would only apply to a dirt road. I suggest "road characteristics" instead (also p. 13, line 360).

*Author response:* Yes, thank you. At line 105, the sentence

*"Road-wear emissions also take into account the soil characteristics, as the pavement hardness"*

is replaced by

*"Road-wear emissions also take into account the road characteristics, as the pavement hardness".*

Also, at line 360, the sentence

*"Note that for road-wear emissions, the soil characteristics used in Bouvelard Alsace Lorraine by Thouron et al. (2018) were employed, leading to higher road-wear emissions than in the EMEP guidelines."*

is replaced by

*"Note that for road-wear emissions, the road characteristics used in Bouvelard Alsace Lorraine by Thouron et al. (2018) were employed, leading to higher road-wear emissions than in the EMEP guidelines.".*

10. p. 5, line 133: "worst quality codes"; I am not sure what the authors mean.

In Sections 2 and 3, it is mentioned that particulate BC is treated using a sectional size distribution with 6 size sections, which is useful to correctly simulate dry deposition. Since atmospheric chemistry is not treated, I assume that aerosol dynamics (condensation/evaporation, nucleation, and coagulation) is also not treated. This seems appropriate as it would have little effect on the BC size distribution (in particular, considering the uncertainties associated with its initial size distribution). The authors should mention this point.

Also, the initial size distribution of emitted particulate BC should be specified, either with a reference or by providing the size distribution.

***Author response:*** The "worst quality codes" mentioned in this study refer to the emission factor quality code indicated in European emission guidelines (Ntziachristos and Samaras, 2016). The quality codes indicated for non-exhaust emissions range from B (emission factors non statistically significant based on a small set of measured re-evaluated data), C (emission factors estimated on the basis of available literature), to D (emission factors estimated applying similarity considerations and/or extrapolation). To improve clarity, at line 132 the sentences

*"Road emission factors vary from 3.8 mg.vkm-1 (Boulter, 2005) to 200 mg.vkm-1 (Thouron et al., 2018), and they present the worst quality codes compared to other wear emission factor. The EMEP guidelines (Ntziachristos and Boulter, 2016) quantify the typical error associated to road wear emission factors to be between 50% to 300%, associated to the difficulties to separate precisely the non-exhaust emission sources and the dependence of soil properties and vehicles operational conditions."*

are replaced by

*"Road emission factors vary from 3.8 mg.vkm-1 (Boulter, 2005) to 200 mg.vkm-1 (Thouron et al., 2018). Compared to other non-exhaust emission sources, road-wear emission factors present the worst quality codes, according to the European emission guidelines EMEP (Ntziachristos and Boulter, 2016), where different quality codes are*

*defined for non-exhaust emissions, ranging from B (emission factors non statistically significant based on a small set of measured re-evaluated data), C (emission factors estimated on the basis of available literature), to D (emission factors estimated applying similarity considerations and/or extrapolation). Road wear is the emission source with the worst quality code, and they highlight the difficulties to separate precisely the non-exhaust emission sources and the dependence of road characteristics and vehicles operational conditions.".*

Also, at line 210, the sentence

*"Because BC is an inert species, this study does not take into account chemical reactions, and only BC concentrations are modelled."*

is followed by

*"The aerosol dynamics (nucleation, coagulation and condensation/evaporation) is also neglected, as it would have no effect on BC mass concentration and low effect on size distribution.".*

The size distribution of non-exhaust particle emissions is specified following the reference of each test. In simulations 2, 4, 5 and 6 followed the size distribution indicated in the European emission guidelines (Ntziachristos and Boulter, 2016). The simulation 3 followed the indications present in the NORTRIP model (Denby et al., 2013a).

These details about the size distribution of emitted particles are included in Section 4.1 (The simulations). At line 355, the sentence

*"Simulations 2 employs the BC wear emission factors indicated in the EMEP guidelines, also used in the HERMES model (Guevara et al., 2019; Ntziachristos and Boulter, 2016)."*

is replaced by

*"Simulation 2 employs the BC wear emission factors and size distribution indicated in*

the EMEP guidelines, also used in the HERMES model (Guevara et al., 2020; Ntzi-
achristos and Boulter, 2016)."

Also, at line 358, the sentence

"Simulation 3 uses the $PM_{10}$ wear emission factors indicated in the NORTRIP model
(Denby et al., 2013a)."

is replaced by

"Simulation 3 uses the $PM_{10}$ wear emission factors and size distribution indicated in
the NORTRIP model (Denby et al., 2013a)."

And finally, at line 363, the sentence

"Simulation 4 employs the same brake and road-wear emission factors as in the EMEP
guidelines, but tyre-wear emission factors are higher."

is followed by

"The same size distribution of non-exhaust emissions indicated in the EMEP guidelines
(Ntziachristos and Boulter, 2016) is employed.".

To improve clarity, an additional Table is included in the Appendix section, with some
more details of the size distribution of emitted particles from non-exhaust emissions.

Please, note that the sentence *"More details about particle size distribution of non-
exhaust emissions are indicated in Table A2, in the Appendix A3."* is included at line
367.

11. p. 10, line 280: it is not clear why $Q_{emis,exh}$ represents both exhaust and non-
exhaust emissions. Please clarify.

***Author response:*** It was corrected.

At line 280 the sentence

*"with $Q_{emis,exh}$ the exhaust and non-exhaust traffic emission rates"*

is replaced by

*"with $Q_{emis,exh}$ the exhaust traffic emission rate"*.

12. p. 10, line 280: it is not clear in $Q_{dep,v}$ what the second subscript, $v$, represents. Please clarify.

*Author response:* The term $v$ indicates the street volume. This is not exactly the same considered in Equation 7 because, in addition to particle deposition over the street surface, it also takes into account the particle deposition over the building surfaces.

To improve clarity, the sentence

$Q_{dep,v}$ *the deposition flux over the street volume, considering the street pavement and building walls surfaces.*

is replaced by

$Q_{dep,v}$ *the deposition flux over the street volume $v$, considering the street pavement and building walls surfaces.*

13. In Section 3, p. 12, Equation 10: please specify the units of EF.

*Author response:* It was added.

14. In Section 4.1, p. 13, line 343: "to estimate and control"; please specify what "control" refers to.

***Author response:*** Some studies propose specific measures to control non-exhaust emissions, such as reducing the limit speed in streets and/or smoothing traffic flow (Begi et al, 2020). We decided to remove this word from the introduction, and it seems more coherent to mention the non-exhaust emission control in the conclusion, and the control was not discussed in the introduction. Then, at line 343, the sentence

*"As mentioned in the introduction, non-exhaust emissions are difficult to estimate and control."*

is replaced by

*"As mentioned in the introduction, non-exhaust emissions are difficult to estimate."*.

And, at line 504, the sentence

*"Following the literature, increasing the BC passenger cars tyre-wear emission factors of the EMEP guidelines from 1.36 mg.vkm$-1$ to 20.8 mg.vkm$-1$ lead to good comparisons of the simulated BC concentrations to the measured ones."*

is followed by

*"Also, more studies are needed to control these emissions. Some studies already indicate the importance of controlling vehicle speed and/or smoothing traffic flow to reduce non-exhaust emissions (Begi et al, 2020). But other aspects can be investigated, such as the road, tyre and brake characteristics, vehicle weight, etc.".*

15. At the end of Section 4.1, the authors mention that their correction factor for exhaust BC emissions used in simulation 6 is based on traffic flow characteristics and BC/PM2.5 uncertainties. Could this be explained in greater detail?

***Author response:*** To improve clarity, the sentences

*"Finally, in order to evaluate the influence of uncertainties in the BC speciation of ex-*

*haust emissions, simulation 6 uses the same non-exhaust emission factors as simulation 4, but BC exhaust emission factors are artificially increased by 23%. This correction factor is defined based on the traffic-flow characteristics observed in the "Boulevard Alsace Lorraine" during the TRAFIPOLLU campaign, and the BC/PM$_{2.5}$ uncertainties for each vehicle class detailed in Table 3-91 of Ntziachristos and Samaras (2018)."*

are replaced by

*"Finally, in order to evaluate the influence of uncertainties in the BC speciation of exhaust emissions, simulation 6 uses the same non-exhaust emission factors as simulation 4, but BC exhaust emissions are artificially increased by 23%. This correction factor is defined using a linear correlation based on $(i)$ the traffic-flow characteristics observed in the "Boulevard Alsace Lorraine" during the TRAFIPOLLU campaign (percentage of diesel and petrol vehicles, according to the vehicle category and technology), and $(ii)$ the BC/PM$_{2.5}$ uncertainties for each vehicle class detailed in Table 3-91 of Ntziachristos and Samaras (2018)."*

16. In Section 4.2, the authors provide two different sets of criteria for model performance evaluation. The stricter set includes 6 performance metrics (5 were initially defined by Hanna et al., Atmos. Environ., 38, 4675-4687, 2004; the normalized absolute difference was added later), whereas the less strict set (suitable for urban areas according to Hanna and Chang, 2012) includes only 4. It is not clear whether the authors consider that the stricter criteria defined for MG and VG should also apply to the less strict set or whether those metrics must simply be dropped from the less strict set; this point should be clarified. It may be a moot point since, in any case, simulations 1 through 3 fail the less strict criteria (except NMSE) and simulations 4 through 6 meet all the stricter criteria.

***Author response:*** In the less strict criteria the MG and VG are not considered. To improve clarity, the sentence

*"Two different criteria are defined, a most strict criteria, with -0.3 < FB < 0.3; 0.7 < MG < 1.3; NMSE < 3; VG < 1.6; FAC2 ≥ 0.5; NAD < 0.3, and a less strict criteria acceptable in urban areas, with -0.67 < FB < 0.67; NMSE < 6; FAC2 ≥ 0.3; NAD < 0.5."*

is replaced by

*"Two different criteria are defined, ($i$) a lest strict criteria, accepted in urban areas, with -0.67 < FB < 0.67; NMSE < 6; FAC2 ≥ 0.3; NAD < 0.5, and ($ii$) a most strict criteria, with -0.3 < FB < 0.3; 0.7 < MG < 1.3; NMSE < 3; VG < 1.6; FAC2 ≥ 0.5; NAD < 0.3.".*

Also, the sentences

*"The BC concentrations observed at "Boulevard Alsace Lorraine" are strongly underestimated in simulations 1, 2 and 3, with a fractional bias (FB) equal to -1.26, -1.10 and -1.15 respectively, not satisfying any performance criterion. The configuration used in simulation 4, with higher tyre and brake-wear emissions, results in satisfactory statistical indicators, respecting all the most strict performance criteria proposed by Hanna and Chang (2012) and Herring and Huq (2018)."*

are replaced by

*"The BC concentrations observed at "Boulevard Alsace Lorraine" are strongly underestimated in simulations 1, 2 and 3, with a fractional bias (FB) equal to -1.26, -1.10 and -1.15 respectively. They do not satisfy any performance criterion, nor the less strict ones, except for normalised mean square error (NMSE). However, the configurations used in simulations 4, 5 and 6, with higher tyre-wear emissions, result in satisfactory statistical indicators. They respect both the less and most strict performance criteria proposed by Hanna and Chang (2012) and Herring and Huq (2018).".*

17. In Section 4.3, p. 17, lines 425-426: I would not say that the observations of Amato

et al. are "quite similar" because the result obtained in this work is slightly above the upper bound of the range given by Amato et al. In Section 4.3, the authors refer to $PM_{10}$ as fine particles. This definition of $PM_{10}$ is unfortunately commonly used in France by some agencies, although it is incorrect. Fine particles correspond to $PM_{2.5}$, and $PM_{10}$, therefore, includes fine particles and a fraction of coarse particles.

***Author response:*** Yes, thank you, the use of the term *fine particles* was corrected. The sentence

*"Therefore, the average mass density of fine particles ($PM_{10}$) in "Boulevard Alsace Lorraine" is about 250 mg.m-2."*

is replaced by

*"Therefore, the average mass density of $PM_{10}$ in "Boulevard Alsace Lorraine" is about 250 mg.m-2."*.

However, both studies (this study and Amato et al.) are focused on $PM_{10}$. Maybe it was not clear before, but please, see the answer of Minor comment 10, that contains the precision about size distribution.

18. In Section 5, p. 19, line 477: "lower concentrations"? Should it be "higher concentrations" if there is double counting?

***Author response:*** Yes, it was corrected, thank you.

The sentence

*"These large differences and the lower concentrations simulated in MUNICH compared to SinG may be explained by the double counting of traffic emissions performed by MUNICH with the one-way coupling technique."*

is replaced by

*"These large differences and the lower concentrations simulated in SinG compared to MUNICH may be explained by the double counting of traffic emissions performed by MUNICH with the one-way coupling technique.".*

19. As mentioned above, the authors should mention the policy implications of their results concerning BC emissions from vehicles in the conclusion.

***Author response:*** As indicated in the specific comment 1, at line 409 the sentence

*"They can be as relevant as exhaust emissions, and their underestimation may justify the virtual increase of BC emissions often employed in street-network modelling studies."*

is followed by

*"This can have direct consequences in public policy to strongly reduce BC concentrations in urban areas: non-exhaust emissions may contribute to as much as half of the BC concentrations in streets.".*

20. Also, some suggestions for additional experimental studies to confirm the results of their work would be appropriate.

***Author response:*** As indicated in the specific comment 1, at line 501 the sentence

*"Following the literature, increasing the BC passenger cars tyre-wear emission factors of the EMEP guidelines from 1.36 mg.vkm$^{-1}$ to 20.8 mg.vkm$^{-1}$ lead to good comparisons of the simulated BC concentrations to the measured ones."*

is followed by

*"The authors highlight that additional work, mostly experimental, should be conducted to confirm this result. However, the work presented here makes a very strong case for*

*revising current BC vehicular emission inventories and taking non-exhaust emissions explicitly into account in air quality simulations."*